

# Detecting and forecasting cryptojacking attack trends in Internet of Things and wireless sensor networks devices

Kishor Kumar Reddy C.[1], Vijaya Sindhoori Kaza[1], Madana Mohana R.[2], Abdulrahman Alamer[3], Shadab Alam[3], Mohammed Shuaib[3], Sultan Basudan[3] and Abdullah Sheneamer[3]

[1] Department of Computer Science, Stanley College of Engineering and Technology for Women, Hyderabad, Telangana, India
[2] Department of Artificial Intelligence and Machine Learning, Chaithanya Bharathi Institute of Technology, Hyderabad, Telangana, India
[3] Department of Computer Science, College of Engineering and Computer Science, Jazan University, Jazan, Gizan, Saudi Arabia

## ABSTRACT

This research addresses the critical issue of cryptojacking attacks, a significant cybersecurity threat where malicious actors covertly exploit computational resources for unauthorized cryptocurrency mining, particularly in wireless sensor networks (WSN) and Internet of Things (IoT) devices. The article proposes an innovative approach that integrates time series analysis with graph neural networks (GNNs) to forecast/detect cryptojacking attack trends within these vulnerable ecosystems. Utilizing the "Cryptojacking Attack Timeseries Dataset," the proposed method emphasizes early detection and predictive insights to anticipate emerging attack patterns. Through rigorous experiments, the model demonstrated high accuracy with ARIMA achieving up to 99.98% on specific attributes and the GNN model yielding an accuracy of 99.99%. Despite these strengths, the ensemble approach showed a slightly lower overall accuracy of 90.97%. Despite the reduction in accuracy compared to individual models, the ensemble method enhances predictive robustness and adaptability, making it more effective in identifying emerging cryptojacking trends amidst varying network conditions. This research significantly contributes to enhancing cybersecurity measures against the evolving threat of cryptojacking in WSN and IoT environments by providing a robust, proactive defence mechanism.

# INTRODUCTION

The rapid production of wireless sensor networks (WSNs) and Internet of Things (IoT) devices have transformed the digital landscape, creating a highly interconnected environment. However, this advancement comes with significant cybersecurity challenges, one of the most pressing being cryptojacking (*Kharraz et al., 2019*; *Carreiro, 2019*). Cryptojacking, the unauthorized use of computer resources to mine cryptocurrencies

Corresponding author
Abdulrahman Alamer,
amalameer@jazanu.edu.sa

without user consent, poses serious risks, particularly in IoT environments where resource constraints and distributed architectures are common (*Androulaki et al., 2018*; *Hong et al., 2018*). The malicious exploitation of devices' central processing unit (CPU) and graphics processing unit (GPU) resources not only degrades system performance but can also lead to increased energy consumption and even physical damage to hardware (*Ali et al., 2020*; *Islam et al., 2020*). Furthermore, cryptojacking is often a precursor to more severe cybersecurity breaches, including malware attacks and data breaches (*Gomes & Correia, 2020*). In the context of WSN and IoT systems, where resource constraints and distributed architectures are common, cryptojacking presents unique risks and challenges. These systems often rely on a multitude of interconnected devices to collect, process, and transmit data, making them vulnerable to exploitation by malicious actors seeking to harness their computational power for illicit cryptocurrency mining (*Gomes & Correia, 2020*; *Lee, Oh & Kim, 2022*). Despite the growing prevalence of cryptojacking attacks in WSN and IoT environments, traditional cybersecurity measures may be ill-equipped to detect and mitigate these threats effectively.

Over the years, various detection methods have been developed to combat cryptojacking. These methods include signature and behaviour-based approaches, machine learning-based detection techniques, network-based monitoring, and blockchain-based solutions (*Abbasi et al., 2023*; *Moreno-Sancho et al., 2023*). For instance, signature and behaviour-based detection methods have been used to identify cryptojacking activities by analysing known patterns and behaviours of malicious scripts (*Eskandari et al., 2018*; *Loose et al., 2023*). Machine learning approaches have been employed to detect anomalies that indicate cryptojacking, while network-based detection has focused on monitoring network traffic for signs of unauthorized cryptocurrency mining (*Kharraz et al., 2019*; *Gomes & Correia, 2020*). Blockchain-based methods, on the other hand, aim to leverage the inherent security features of blockchain technology to detect and prevent cryptojacking activities (*Androulaki et al., 2018*; *The Telegraph, 2018*).

Despite the progress made by these existing methods, they each come with significant limitations. Signature and behaviour-based detection techniques struggle to keep up with the rapidly evolving nature of cryptojacking tactics, making them less effective against new and unknown threats (*Abbasi et al., 2023*; *Eskandari et al., 2018*). Machine learning-based approaches, while powerful, often suffer from high false-positive rates, especially in environments with high legitimate workloads (*Gomes & Correia, 2020*; *Naseem et al., 2021*). Network-based detection methods can be limited by the complexity and distribution of modern IoT networks, where monitoring all network traffic can be challenging (*Muñoz, Suárez-Varela & Barlet-Ros, 2019*; *Pott, Gulmezoglu & Eisenbarth, 2023*). Blockchain-based detection, while promising, requires further validation and adaptation to be effective in real-world scenarios, particularly in resource-constrained IoT environments (*Androulaki et al., 2018*; *The Telegraph, 2018*; *Rajasoundaran et al., 2021*).

Given the limitations of existing methods, there is a clear need for a more robust and proactive approach to cryptojacking detection in IoT and WSN environments. This article proposes a novel methodology that combines time series analysis with graph neural networks (GNNs) to detect and forecast cryptojacking attack trends. By predicting

potential cryptojacking activities before they fully take place, this approach aims to provide early warnings and enable timely countermeasures. The objective is to enhance the overall security of IoT and WSN systems by reducing the impact of cryptojacking on device performance, energy consumption, and system reliability. The novelty of this research lies in its integration of time series modelling with GNNs to not only detect but also predict cryptojacking trends in IoT environments. Unlike traditional detection methods that focus on identifying ongoing or past cryptojacking activities, this approach seeks to anticipate future attacks, thereby offering a proactive defence mechanism. This predictive capability is particularly valuable in the context of IoT and WSN systems, where the ability to forecast potential threats can significantly mitigate their impact.

The major contributions of this article are as follows:

- **Development of a forecasting model:** The article introduces a forecasting model that leverages time series analysis and GNNs to predict cryptojacking attack trends in IoT and WSN environments.
- **Comprehensive evaluation:** The proposed model is evaluated using a real-world dataset, the "Cryptojacking Attack Timeseries Dataset," demonstrating its effectiveness in predicting and mitigating cryptojacking threats.
- **Proactive defence strategy:** By providing early detection and predictive insights, the proposed method offers a proactive defence strategy that enhances the security of IoT and WSN systems against cryptojacking.

The rest of the article is organized as follows: Section "Literature Survey" provides a detailed literature survey, reviewing existing methods for cryptojacking detection and their limitations. Section "Materials and Methods" describes the proposed methodology, including the data collection, preprocessing, data analysis, and feature selection. Section "Proposed Work" presents the model development and evaluation processes. Section "Results" discusses the experimental results and implications of the findings. Finally, Section "Conclusion" concludes the article by summarizing the key contributions and the significance of the proposed approach, it also gives insights into future directions that can be explored.

## Literature survey

*Abbasi et al. (2023)* introduced a hybrid detection approach that combines signature-based and behavior-based methods to detect cryptojacking malware in real-time. Their technique was specifically designed to combat in-browser cryptojacking by identifying and blocking known malicious scripts while also monitoring abnormal behavior patterns indicative of cryptojacking. While this method has proven effective in dealing with known threats, it faces significant challenges in maintaining accuracy due to the rapid evolution of cryptojacking techniques, which often bypass traditional detection mechanisms by continuously altering their signature and behavior profiles.

*Eskandari et al. (2018)* also proposed a hybrid detection system focused on browser-based cryptojacking. Their approach provided valuable insights into the challenges

associated with detecting stealthy cryptojacking activities, particularly those that obscure their presence by minimizing CPU usage or spreading their operations across multiple devices. Although their method offers a comprehensive detection framework, it struggles with the inherent difficulty of identifying covert mining activities, which often operate below the detection thresholds of traditional monitoring tools.

*Kharraz et al. (2019)* developed the Outguard method, a machine learning-based approach aimed at detecting in-browser covert cryptocurrency mining activities. This method leverages a variety of machine learning algorithms to analyze and identify patterns associated with cryptojacking. However, despite its innovative use of machine learning, the approach faces challenges in accurately detecting stealthy mining operations, particularly those that have been optimized to evade traditional anomaly detection techniques.

*Gomes & Correia (2020)* proposed another machine learning-based detection method that relies on CPU usage metrics to identify cryptojacking activities. By monitoring abnormal spikes in CPU usage, their system can detect when a device is being exploited for unauthorized cryptocurrency mining. However, this approach is prone to generating false positives, especially in environments where legitimate applications may also cause high CPU utilization, thus complicating the differentiation between normal and malicious activities.

*Muñoz, Suárez-Varela & Barlet-Ros (2019)* explored a network-based approach to cryptojacking detection using NetFlow/IPFIX network measurements. Their method focuses on identifying unusual patterns in network traffic that are indicative of cryptojacking activities, such as sustained high outbound data volumes or frequent connections to cryptocurrency mining pools. While this technique is effective in simpler network environments, it encounters limitations in more complex and highly distributed IoT networks, where the sheer volume of data and the complexity of the network topology can hinder accurate detection.

*Androulaki et al. (2018)* and *Gilad et al. (2017)* investigated blockchain-based detection methods, leveraging the decentralized and transparent nature of blockchain technology to enhance cryptojacking detection. *Androulaki et al. (2018)* utilized Hyperledger Fabric, a permissioned blockchain platform, to improve the traceability and accountability of cryptojacking activities. Similarly, *Gilad et al. (2017)* proposed using Algorand, a scalable blockchain protocol, to enhance the security and efficiency of cryptojacking detection. While these blockchain-based methods show promise in providing a secure and reliable detection framework, their effectiveness is highly dependent on the specific use cases and requires further validation and adaptation to be applicable in diverse environments, particularly those involving IoT devices.

Despite the progress made by the existing methods, they each exhibit significant limitations that hinder their effectiveness in combating cryptojacking, particularly in IoT and WSN environments. Hybrid methods, while comprehensive, are often outpaced by the rapid evolution of cryptojacking techniques, leading to reduced accuracy over time (*Romano, Zheng & Wang, 2020*). Machine learning-based approaches, although powerful, tend to produce high false-positive rates in environments with heavy legitimate workloads, making them less reliable in practical applications (*Sachan, Agarwal & Shukla, 2022*;

**Table 1 Summary and Insights obtained from existing literature.**

| References | Methods | Techniques | Results | Problems identified |
|---|---|---|---|---|
| *Abbasi et al. (2023)* | Hybrid (Signature-based & behavior-based) | Real-time approach | Combating in-browser cryptojacking malware | Accuracy challenges in rapidly evolving techniques |
| *Eskandari et al. (2018)* | Hybrid (Signature-based & behavior-based) | Browser-based detection | Insight into challenges of detecting stealthy mining | Difficulty in detecting covert mining activities |
| *Kharraz et al. (2019)* | Machine learning-based | Outguard method | In-browser covert cryptocurrency mining detection | Challenges in detecting stealthy mining |
| *Gomes & Correia (2020)* | Machine learning-based | CPU usage metric-based detection | Detection of cryptojacking activities | Potential false positives under heavy legitimate workloads |
| *Muñoz, Suárez-Varela & Barlet-Ros (2019)* | Network-based | NetFlow/IPFIX network measurements | Detection of cryptocurrency miners, including cryptojacking | Limitations in highly distributed and complex environments |
| *Androulaki et al. (2018)* | Blockchain-based | Hyperledger fabric | Potential enhancement of blockchain-based cryptojacking detection | Applicability variations based on use cases and validation |
| *Gilad et al. (2017)* | Blockchain-based | Algorand | Potential enhancement of blockchain-based cryptojacking detection | Applicability variations based on use cases and validation |

*Xu et al., 2022*). Network-based detection methods are limited by the complexity of modern networks, particularly in highly distributed IoT systems where monitoring and analysing all network traffic becomes challenging (*Singh et al., 2020*). Lastly, blockchain-based detection methods, though innovative, require further adaptation and validation to be effective across different scenarios and use cases, particularly given the resource constraints of IoT devices.

The proposed model in this article seeks to address these limitations by integrating time series analysis with GNNs to not only detect but also forecast cryptojacking attacks. This novel approach shifts the focus from reactive to proactive defence, enabling the prediction of cryptojacking trends before they manifest fully. By analysing temporal patterns in cryptojacking activities and leveraging the relational data modelling capabilities of GNNs, this model can anticipate potential threats, thus providing early warnings and allowing for timely interventions. Additionally, the use of ensemble methods enhances the robustness and accuracy of the predictions, reducing the likelihood of false positives and improving detection in complex IoT and WSN environments. This approach, therefore, not only overcomes the demerits of existing methods but also introduces a new standard in cryptojacking detection, one that is more suited to the dynamic and resource-constrained nature of IoT systems. With a focus on wireless sensor networks and Internet of Things environments, this research introduces a novel method to detect and mitigate cryptojacking, addressing the constraints and insights presented in Table 1, which provides a summary of various cryptojacking detection methods and their effectiveness, as reported in existing literature. This table categorizes the methods into different types such as hybrid, machine learning-based, network-based, and blockchain-based approaches.

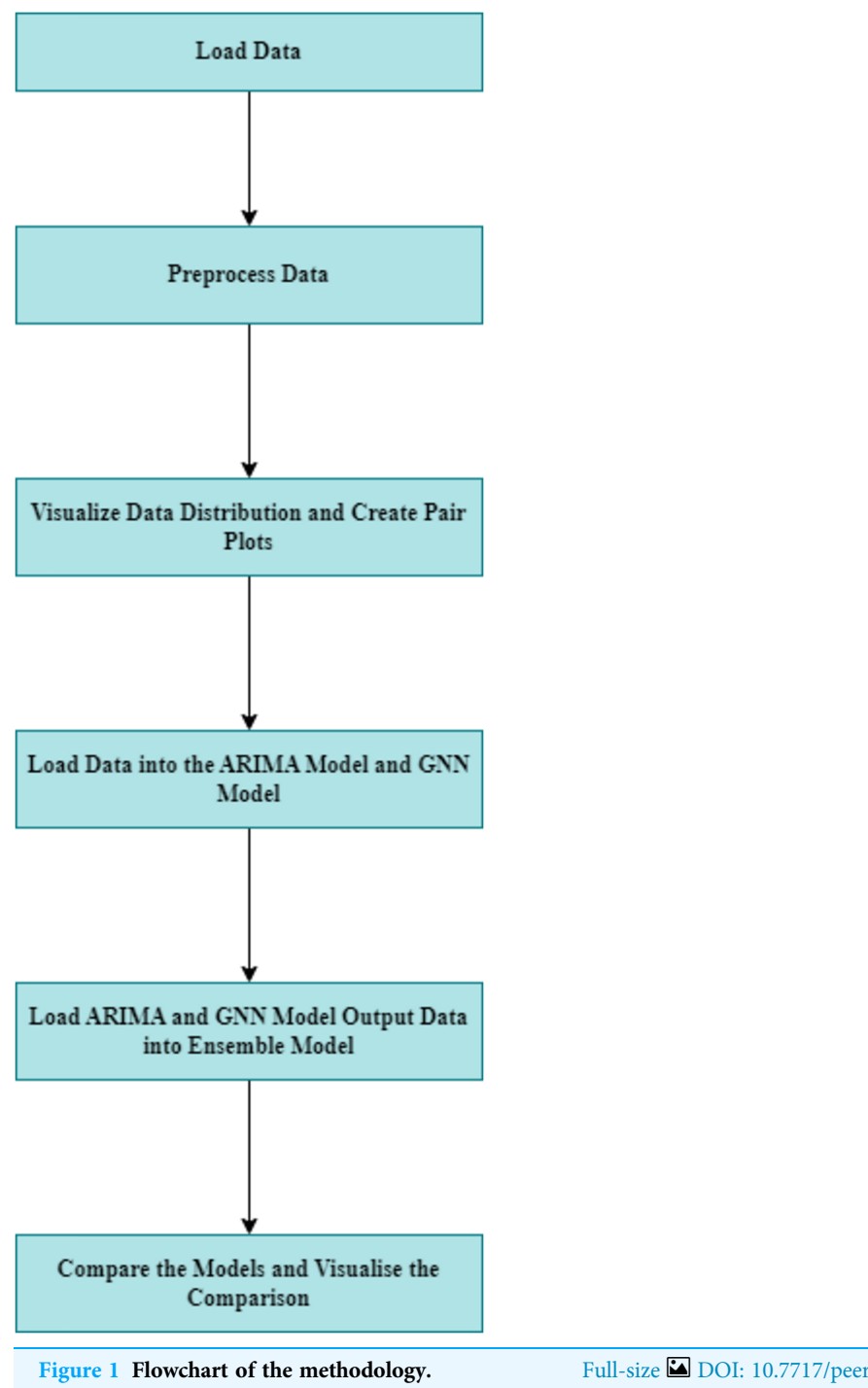

**Figure 1 Flowchart of the methodology.**

## MATERIALS AND METHODS

Figure 1 gives a overview of the steps and flow of the process followed in ordered to obtain the said results. To replicate the model further subsections and Algorithm 1 give details of the steps and methodology.

**Algorithm 1** Algorithm for data loading, preprocessing, exploratory data analysis

***Step 1: Loading and Preprocessing Data***

*Input:* CSV files: 'final-complete-data-set.csv', 'final-anormal-data-set.csv', and 'final-normal-data-set.csv'

*Output:* Preprocessed DataFrame 'df'

*1. Load data from CSV files into pandas DataFrames:*

   *df <- load_data('final-complete-data-set.csv')*

   *df_a <- load_data('final-anormal-data-set.csv')*

   *df_n <- load_data('final-normal-data-set.csv')*

*2. Concatenate 'df_a' and 'df_n' DataFrames along axis 0:*

   *df <- concatenate_dataframes(df_a, df_n)*

*3. Handle missing values:*

   *list15 <- identify_missing_values_over_15_percent(df)*

   *df <- remove_columns_with_missing_values(df, list15)*

   *list0 <- identify_remaining_missing_values(df)*

   *df <- drop_rows_with_missing_values(df, list0)*

*4. Classify variables into categories:*

   *df <- vardefiner(df)*

*5. Filter columns:*

   *df <- drop_columns_without_information(df)*

   *df <- remove_single_valued_categorical_variables(df)*

*6. Convert categorical variables to float type:*

   *df <- convert_categorical_to_float(df)*

*7. Additional Data Preprocessing:*

   *df <- divide_numeric_variables_into_categories(df)*

   *df <- create_binary_flags_for_numeric_variables(df)*

   *df <- compute_total_flags_for_each_record(df)*

   *df <- create_binary_variable_for_cryptojacking_risk(df)*

***Step 2: Exploratory Data Analysis (EDA)***

*Input:* Preprocessed DataFrame 'df'

*1. Visualize data distribution:*

   *visualize_data_distribution(df)*

*2. Create pairplot:*

   *create_pairplot(df)*

***Step 3: Time Series Analysis***

*Input*: Preprocessed DataFrame 'df'

*Output:* ARIMA models and forecast accuracy

*1. Fit ARIMA models to selected time series attributes:*

   *arima_models <- fit_arima_models(df)*

(Continued)

| Algorithm 1 (continued) |
|---|

*2. Evaluate ARIMA forecast accuracy on the test set:*

   *arima_accuracy <- evaluate_arima_forecast_accuracy(arima_models)*

***Step 4: Graph Neural Network (GNN) Model***

*Input:* Preprocessed DataFrame 'df'

*Output:* Trained GNN model and evaluation metrics

*1. Prepare data for GNN model:*

   *data <- prepare_data_for_gnn(df)*

*2. Define GNN model:*

   *gnn_model <- define_gnn_model()*

*3. Train GNN model:*

   *trained_gnn_model <- train_gnn_model(gnn_model, data)*

*4. Evaluate GNN model on the test set:*

   *gnn_accuracy <- evaluate_gnn_model(trained_gnn_model, data)*

***Step 5: Ensemble Model***

*Input:* ARIMA models, trained GNN model

*Output:* Ensemble model and evaluation metrics

*1. Combine ARIMA and GNN predictions to create the ensemble model:*

   *ensemble_model <- create_ensemble_model(arima_models, trained_gnn_model)*

*2. Evaluate ensemble model performance:*

   *ensemble_accuracy <- evaluate_ensemble_model(ensemble_model)*

***Step 6: Model Comparison Visualization***

*Input:* ARIMA accuracy, GNN accuracy, Ensemble accuracy

*Output:* Visualization comparing model accuracies

*1. Compare accuracy of ARIMA, GNN, and Ensemble models for each attribute:*

   *compare_model_accuracies(arima_accuracy, gnn_accuracy, ensemble_accuracy)*

*2. Compare overall accuracies of the three models using a bar chart:*

   *visualize_overall_model_accuracies(arima_accuracy,gnn_accuracy, ensemble_accuracy)*

## Description of the cryptojacking attack timeseries dataset

A useful and specialized source of time series data, the "Cryptojacking Attack Timeseries Dataset" is intended to study the performance of WSN and IoT server instances during cryptojacking attacks and support research and development in real-time detection of the attacks. This dataset includes comprehensive data on metrics for server performance that were gathered during mock cryptojacking attacks. Numerous characteristics pertaining to CPU utilization (Tables 2–4), other performance indicators (Table 5), memory usage (Table 6), and network activity (Table 7), are included. Because every entry in the dataset has a date, it is possible to view performance metrics during the simulated attacks in

**Table 2 Summary statistics when load on CPU core is greater than 1.**

|       | load_cpucore | load_min1 | load_min15 | load_min5 |
|-------|--------------|-----------|------------|-----------|
| count | 4,309        | 4,309     | 4,309      | 4,309     |
| mean  | 2            | 3.680733  | 3.135024   | 3.545523  |
| std   | 0            | 0.412826  | 0.899465   | 0.621365  |
| min   | 2            | 0.96      | 0.18       | 0.42      |
| 25%   | 2            | 3.6       | 2.89       | 3.67      |
| 50%   | 2            | 3.73      | 3.6        | 3.74      |
| 75%   | 2            | 3.86      | 3.7        | 3.79      |
| max   | 2            | 4.65      | 3.8        | 4         |

**Table 3 Summary statistics when load on CPU core is less than or equal to 1.**

|       | load_cpucore | load_min1 | load_min15 | load_min5 |
|-------|--------------|-----------|------------|-----------|
| count | 91,001       | 91,001    | 91,001     | 91,001    |
| mean  | 1            | 1.333823  | 1.26414    | 1.315291  |
| std   | 0            | 1.412298  | 1.334691   | 1.380397  |
| min   | 1            | 0         | 0.05       | 0.02      |
| 25%   | 1            | 0.17      | 0.2        | 0.19      |
| 50%   | 1            | 0.41      | 0.25       | 0.3       |
| 75%   | 1            | 2.74      | 2.55       | 2.96      |
| max   | 1            | 5.55      | 4.2        | 4.51      |

**Table 4 Summary statistics of CPU feature variables.**

|       | cpu_idle | cpu_iowait | cpu_nice | cpu_softirq | cpu_system | cpu_total | cpu_user | percpu_0_idle | percpu_0_iowait | percpu_0_nice | percpu_0_softirq | percpu_0_system | percpu_0_total | percpu_0_user |
|-------|----------|-----------|----------|-------------|------------|-----------|----------|---------------|-----------------|---------------|------------------|-----------------|----------------|---------------|
| count | 95,310   | 95,310    | 95,310   | 95,310      | 95,310     | 95,310    | 95,310   | 95,310        | 95,310          | 95,310        | 95,310           | 95,310          | 95,310         | 95,310        |
| mean  | 50.02    | 0.003     | 0.004    | 0.069       | 3.349      | 49.93     | 39.91    | 50.05         | 0.004           | 0.004         | 0.055            | 3.318           | 49.95          | 39.12         |
| std   | 45.2     | 0.05      | 0.46     | 0.27        | 1.52       | 45.2      | 41.9     | 45.2          | 0.06            | 0.46          | 0.23             | 1.55            | 45.3           | 41.0          |
| min   | 0        | 0         | 0        | 0           | 0          | 3.2       | 1        | 0             | 0               | 0             | 0                | 0               | 3.3            | 1             |
| 25%   | 0        | 0         | 0        | 0           | 2.5        | 8.3       | 5.5      | 0             | 0               | 0             | 0                | 2.5             | 8.3            | 5.5           |
| 50%   | 88.9     | 0         | 0        | 0           | 2.8        | 11.1      | 7.3      | 88.9          | 0               | 0             | 0                | 2.8             | 11.1           | 7.3           |
| 75%   | 91.7     | 0         | 0        | 0           | 3.7        | 100       | 95.2     | 91.7          | 0               | 0             | 0                | 3.7             | 100            | 94.5          |
| max   | 94.3     | 3.6       | 75.5     | 3           | 48         | 100       | 99.2     | 96.7          | 10              | 74.5          | 3                | 26.8            | 100            | 100           |

chronological order and examine the changing trends and features of cryptojacking attacks over time by using this temporal data.

Table 2 offers insights into the CPU utilization metrics during periods when the load on the CPU core exceeded 1. It summarizes key performance indicators such as load_cpucore, which counts the instances when the CPU load was above 1, and load_min1, load_min5, and load_min15, which show the average CPU load over 1, 5, and 15 min, respectively. The

**Table 5  Summary statistics of diskio feature variables.**

| | diskio_sda1_write_bytes | diskio_sda_write_bytes | diskio_sda1_read_bytes | diskio_sda1_time_since_update | diskio_sda_read_bytes | diskio_sda_time_since_update |
|---|---|---|---|---|---|---|
| count | 9.53 | 9.53 | 95,310 | 95,310 | 9.53 | 95,310 |
| mean | 1.93 | 2.34 | 6.403357 | 1.162641 | 2.19 | 1.162641 |
| std | 1.21 | 1.89 | 1,481.39172 | 0.121199 | 1.38 | 0.121199 |
| min | 0 | 0 | 0 | 0.6527 | 0 | 0.6527 |
| 25% | 0 | 0 | 0 | 1.087129 | 0 | 1.087129 |
| 50% | 0 | 0 | 0 | 1.093017 | 0 | 1.093017 |
| 75% | 0 | 0 | 0 | 1.209975 | 0 | 1.209975 |
| max | 3.05 | 3.47 | 454,656 | 2.127072 | 1.92 | 2.127072 |

**Table 6  Summary statistics of memory feature variables.**

| | mem_active | mem_available | mem_cached | mem_free | mem_inactive | mem_used | memswap_free | memswap_sin | memswap_sout | memswap_total | memswap_used | mem_buffers | mem_percent | mem_shared | mem_total | memswap_percent |
|---|---|---|---|---|---|---|---|---|---|---|---|---|---|---|---|---|
| count | 9.53 | 9.53 | 9.53 | 9.53 | 9.53 | 9.53 | 9.53 | 95,310 | 9.53 | 9.53 | 9.53 | 9.53 | 95,310 | 9.53 | 9.53 | 95,310 |
| mean | 7.30 | 2.64 | 1.13 | 2.64 | 4.68 | 1.01 | 9.79 | 1,407.62 | 8.79 | 9.80 | 9.74 | 1.52 | 28.04 | 8.84 | 3.66 | 0.0133 |
| std | 4.36 | 8.22 | 5.76 | 8.22 | 2.60 | 8.03 | 2.31 | 10,478.8 | 7.90 | 2.32 | 8.04 | 9.43 | 20.7 | 3.82 | 4.77 | 0.12 |
| min | 1.99 | 3.81 | 3.62 | 3.81 | 1.71 | 4.58 | 0 | 0 | 0 | 0 | 0 | 0 | 11.5 | 6.94 | 1.04 | 0 |
| 25% | 5.27 | 2.58 | 7.17 | 2.58 | 3.33 | 6.61 | 0 | 0 | 0 | 0 | 0 | 1.68 | 17.8 | 8.86 | 3.71 | 0 |
| 50% | 5.65 | 3.03 | 8.78 | 3.03 | 3.33 | 6.78 | 0 | 0 | 0 | 0 | 0 | 2.12 | 18.3 | 8.90 | 3.71 | 0 |
| 75% | 6.28 | 3.05 | 1.22 | 3.05 | 5.46 | 8.82 | 0 | 0 | 0 | 0 | 0 | 2.12 | 30.3 | 8.96 | 3.71 | 0 |
| max | 2.04 | 3.52 | 2.23 | 3.52 | 9.44 | 3.59 | 6.46 | 98,304 | 7.22 | 6.46 | 7.34 | 2.16 | 90.4 | 9.07 | 3.97 | 1.1 |

**Table 7  Summary statistics of network feature variables.**

| | network_lo_cumulative_cx | network_lo_cumulative_rx | network_lo_cumulative_tx | network_lo_cx | network_lo_rx | network_lo_time_since_update | network_lo_tx |
|---|---|---|---|---|---|---|---|
| count | 95,310 | 95,310 | 95,310 | 95,310 | 95,310 | 95,310 | 95,310 |
| mean | 31,780.51206 | 15,890.256 | 15,890.256 | 2.28039 | 1.140195 | 1.162641 | 1.140195 |
| std | 42,494.9283 | 21,247.464 | 21,247.464 | 25.568 | 12.784 | 0.121031 | 12.784 |
| min | 400 | 200 | 200 | 0 | 0 | 0.69546 | 0 |
| 25% | 12,952 | 6,476 | 6,476 | 0 | 0 | 1.087103 | 0 |
| 50% | 18,464 | 9,232 | 9,232 | 0 | 0 | 1.092982 | 0 |
| 75% | 19,940 | 9,970 | 9,970 | 0 | 0 | 1.209952 | 0 |
| max | 194,680 | 97,340 | 97,340 | 656 | 328 | 2.114878 | 328 |

data provides an understanding of how the CPU behaves under higher stress conditions, revealing trends and patterns that could be indicative of cryptojacking activities. For instance, the average load over 1 min might spike during an attack, signaling abnormal activity that warrants further investigation.

Table 3 presents similar metrics as Table 2 but focuses on periods when the CPU load was less than or equal to 1. This table is crucial for comparing CPU performance under

normal conditions *versus* high-load conditions (as in Table 2). The statistics include averages, standard deviations, and ranges for the CPU load metrics. For example, the average CPU load (load_min1) is lower, with a smaller standard deviation, indicating more stable performance during these periods. Analyzing these statistics helps in understanding the baseline CPU behavior, which can be contrasted against high-load periods to detect anomalies associated with cryptojacking.

Table 4, displays a range of statistics pertaining to different CPU attributes that were obtained from the "Cryptojacking Attack Timeseries Dataset." This table includes several CPU-related parameters that provide insights into the distribution of these performance metrics, including:

- Counts for every measure, showing how many occurrences of each CPU feature have been seen.
- The central tendency and data distribution for each feature are provided by the means and standard deviations.
- The range of numbers that indicate the minimum and maximum operating parameters of various CPU functions.
- Quartile values reflect the median (50th percentile), as well as the lower and upper quartiles of the data distribution.

Table 4 details various CPU feature statistics, offering a comprehensive look at the CPU's performance characteristics over time. This table is crucial for identifying normal operating ranges and detecting deviations that might suggest a cryptojacking attack.

Table 5 provides statistical insights into disk I/O operations, similar to how Table 4 addresses CPU features. This table helps to understand how disk activity is impacted during cryptojacking attacks, as abnormal I/O patterns can be a sign of such malicious activity. This table includes:

- Counts for every measure, which show how many instances of each disc I/O feature have been recorded.
- Means and standard deviations, which show each feature's central tendency and data distribution.
- Minimum and maximum values that indicate the bounds on which certain disc I/O functions can be used.
- Quartile numbers showing the median (50th percentile) and the lower and upper quartiles of the data distribution.

Table 6 focuses on memory-related metrics, summarizing the distribution and central tendencies of memory usage during the dataset's period. By analyzing counts, means, standard deviations, and range values, this table helps in understanding how memory usage fluctuates during normal operation and potential cryptojacking attacks. High memory usage or unusual memory patterns could indicate the presence of such attacks. This table provides the following statistics:

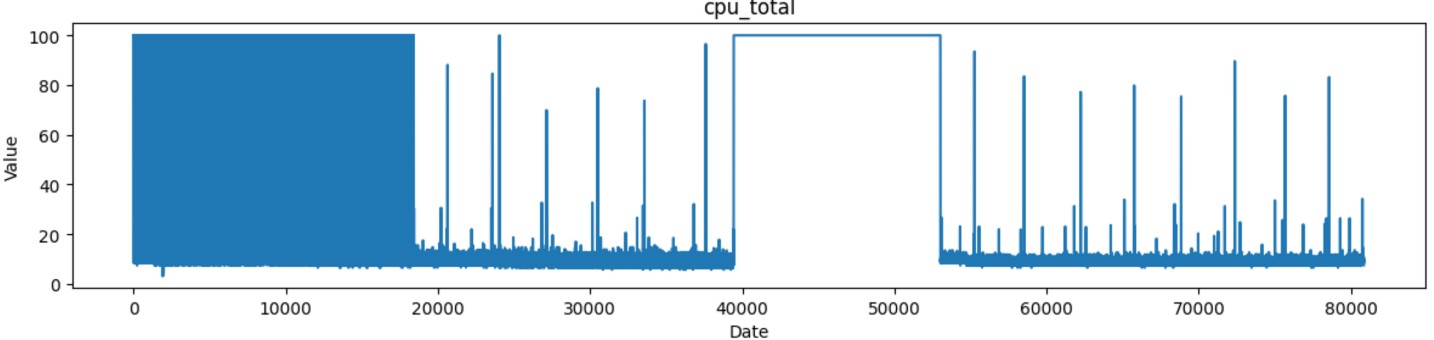

**Figure 2  Plot of values of CPU_total attribute in the dataset.**     

- Counts for each metric, which indicate how many occurrences of each memory-related trait have been recorded.
- Means and standard deviations, which shed light on each feature's data distribution and central tendency.
- The lowest and maximum values that indicate the range in which these memory-related functions function.
- Quartile numbers illustrating the data distribution; these include the median (50th percentile) and the lower and upper quartiles.

Table 7 offers a statistical analysis of network-related metrics, providing counts, averages, standard deviations, minimum and maximum values, and quartile values for various network performance indicators. This table is particularly important for understanding how network activity changes during cryptojacking attacks. Abnormal increases in network traffic, unusual patterns, or spikes in specific metrics could be indicative of unauthorized cryptocurrency mining activities, which rely on network resources to communicate with command-and-control servers or mining pools.

By providing detailed statistical summaries of CPU, memory, disk I/O, and network features, these tables collectively help in identifying patterns and anomalies that may indicate cryptojacking attacks, offering a comprehensive view of the system's performance during such incidents.

The term "CPU_total" refers to a column that holds information on the overall CPU load or usage, which shows how much of the CPU is used overall in a system or computer. As shown in Fig. 2, CPU use is a fundamental indicator used to track the workload and performance of a computer's CPU. Keeping an eye on this is crucial for a number of reasons, including spotting possible performance bottlenecks, figuring out whether the CPU is being overworked or underutilized, and discovering problems with the responsiveness of the system.

As shown in Fig. 3, the CPU load average over a brief time window—typically one minute—is represented by the 'load min1' column. CPU load is a measurement of the

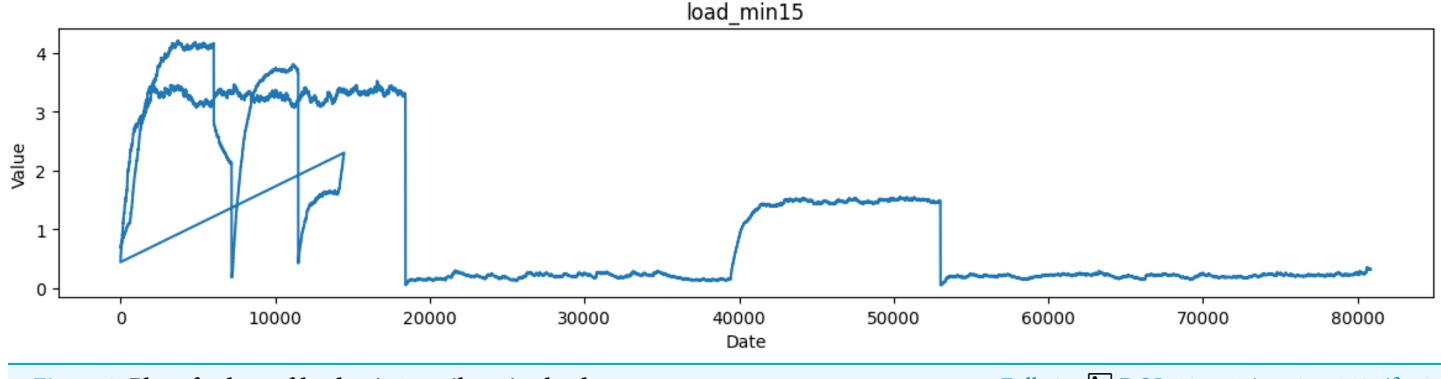

**Figure 3 Plot of values of load_min1 attribute in the dataset.**

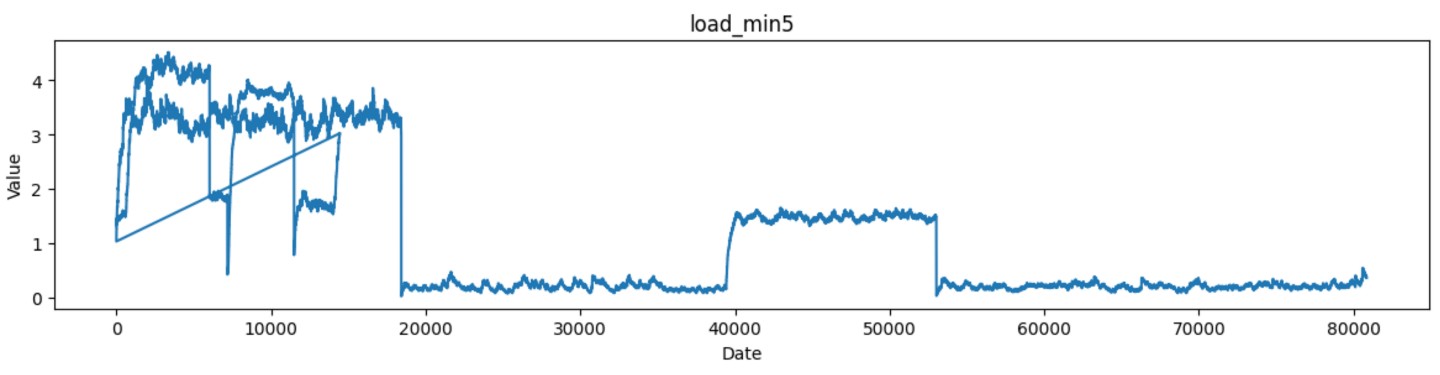

**Figure 4 Plot of values of load_min5 attribute in the dataset.**

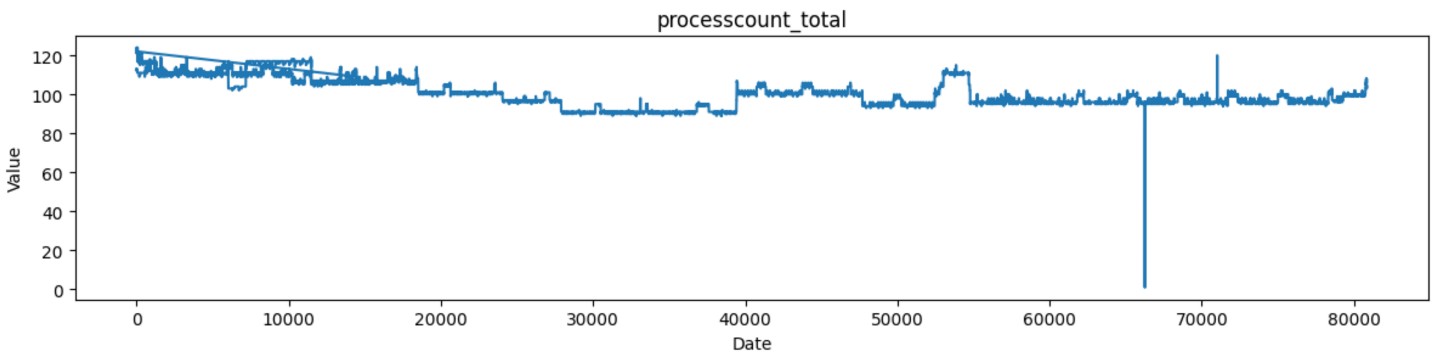

**Figure 5 Plot of values of load_min15 attribute in the dataset.**

amount of processing work the CPU is doing at any one time. It displays the quantity of processes that are awaiting execution in the system's queue.

The 'load_min5' column represents the CPU load average over a medium time window, typically five minutes as depicted in Fig. 4. As with the other load columns, it shows the average number of processes waiting to be executed over a 5-min period.

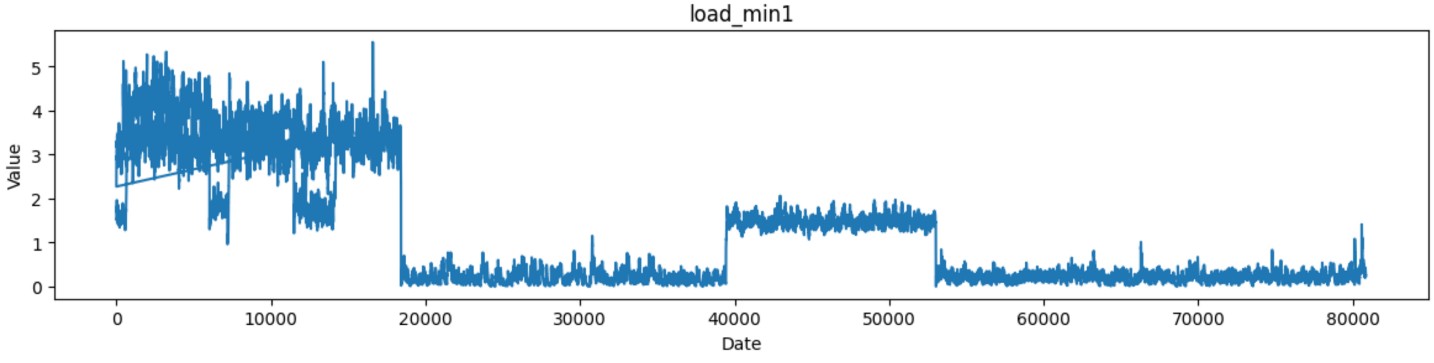

**Figure 6** **Plot of values of processcount_running attribute in the dataset.**

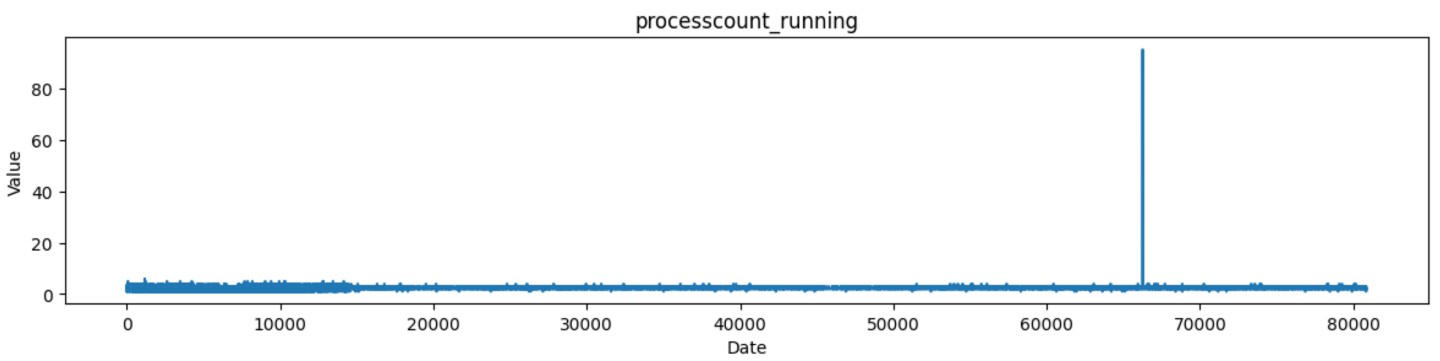

**Figure 7** **Plot of values of processcount_total attribute in the dataset.**

The 'load_min15' column represents the CPU load average over a longer time window, typically fifteen minutes as depicted in Fig. 5. Like load_min1, it measures the average number of processes waiting for execution over a 15-min period. These load averages are crucial metrics for system administrators and performance monitoring because they help in assessing the system's load and performance. A low load value usually indicates that the system is handling its tasks well and not overloaded, while a high load value may indicate resource contention and possible performance issues.

The 'Processcount_running' column likely represents the count of processes that are currently running on the system at a specific moment as depicted in Fig. 6. A running process is a program or task that is currently executing and actively using system resources, such as CPU and memory.

Monitoring the number of running processes is useful for understanding the current workload on the system and assessing its real-time activity. Based on Fig. 7, the "Processcount_total" column most likely indicates the total count of processes on the system, including both operating and non-running (such as sleeping or waiting) processes. It shows the total number of applications or tasks that have been launched or started on the system since its initial boot.

The availability of a well-designed dataset of this kind is essential for testing and comparing novel detection methods and algorithms. By utilizing the performance metrics recorded in this dataset, researchers can develop real-time detection methods with low computational overhead, addressing one of the primary concerns in cryptojacking defense.

## Data collection and preprocessing

The dataset utilized in this research comprises three main subsets: final-complete-data-set.csv, final-anormal-data-set.csv, and final-normal-data-set.csv. These datasets were collected from WSN and IoT systems and contain various system attributes relevant to detecting cryptojacking activity. To prepare the data for analysis, several preprocessing steps were performed. Initially, missing values were addressed by removing rows with null values exceeding 15% of the total observations. Subsequently, columns with any remaining missing values were dropped to ensure data integrity.

Furthermore, categorical variables were converted to numeric format to facilitate analysis. Variables were categorized as binary, categorical, float, or integer based on their data types, and irrelevant or redundant features were removed to streamline the dataset. Notably, certain variables related to system resources such as CPU, memory, disk I/O, network activity, file system usage, and process counts were identified and grouped for further analysis. Additionally, flagging mechanisms were implemented to identify anomalous behavior within these resource-related variables, aiding in the detection of potential cryptojacking activity. The flagged records were then analyzed to determine the total number of flags associated with each observation, with a higher flag count indicating a greater likelihood of cryptojacking. Subsequently, a predictive model was developed using the KMeans clustering algorithm based on selected features (fs_/_free, mem_cached, and memswap_free) to further classify potential cryptojacking instances. The steps are described in Algorithm 1.

## Exploratory data analysis

Following data preprocessing, exploratory data analysis (EDA) was conducted to gain insights into the distribution and relationships among the selected features. Histograms and pair plots were utilized to visualize the distribution of each feature and explore potential correlations between them. Moreover, the SARIMAX model was employed to analyze the time series data, with the negative log likelihood value indicating the model's suitability for capturing temporal patterns and trends within the dataset. The steps are described in Algorithm 1.

## Feature selection process

Feature selection is a critical step in the development of any machine learning model, particularly in the context of cryptojacking detection in IoT and WSN environments. The selection of appropriate features not only improves the model's performance but also reduces computational complexity, which is essential in resource-constrained environments like IoT networks (*Carlin et al., 2019*).

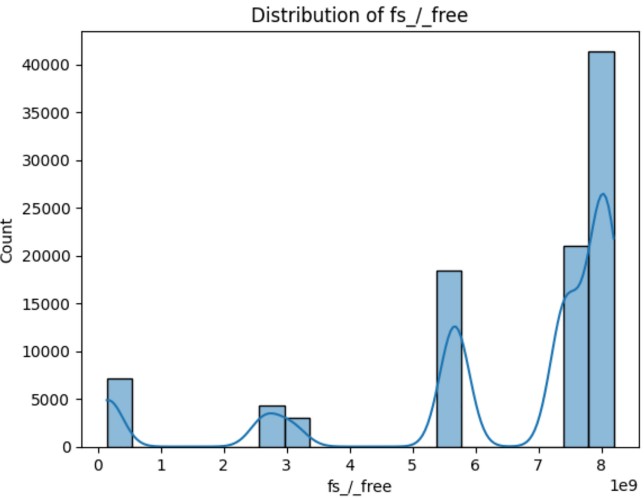

**Figure 8** Distribution of fs_/_free (available space in this filesystem).

First, the potential features were identified by reviewing existing research on cryptojacking detection, cybersecurity, and IoT networks. These features were then categorized into groups like network traffic, device resource usage, and patterns associated with cryptojacking. They were then pre-processed and engineered, missing values were handled, data was normalized, and new features were created to capture more complex patterns. To avoid redundancy, a correlation analysis was conducted and the less important features were discarded.

These features were then ranked using algorithms like Random Forest. Features that consistently ranked high were retained, while those with little predictive power were excluded. To further improve efficiency, dimensionality reduction techniques like principal component analysis (PCA) were applied. Finally, the selected features were validated using cross-validation to ensure their effectiveness in real-world scenarios. Through this rigorous process, the key indicators of cryptojacking, such as sudden spikes in CPU usage, abnormal network traffic patterns, and specific temporal behaviours, were identified. These features were then used as inputs to the proposed model, enabling effective cryptojacking detection in IoT and WSN environments. By carefully selecting and refining the features, the proposed model was able to achieve a balance between accuracy and computational efficiency, making it well-suited for deployment in real-world IoT systems where resources are often constrained. This rigorous feature selection process was instrumental in enhancing the overall performance of the cryptojacking detection model, ensuring that it could operate effectively in diverse and dynamic environments.

## Overview regarding selected attributes

The three attributes fs_/_free, mem_cached, and memswap_free themselves do not directly indicate cryptojacking in WSN and IoT (*Ali et al., 2020*). However, they can be relevant in the context of detecting cryptojacking or any suspicious activities related to it

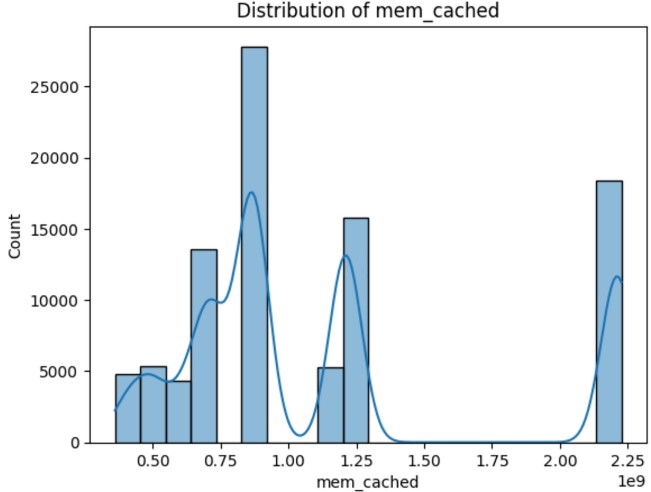

**Figure 9  Distribution of mem_cached (amount of memory used for caching).**

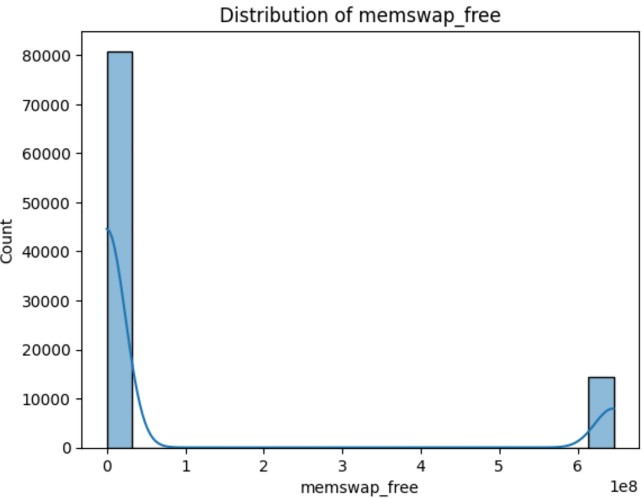

**Figure 10  Distribution of memswap_free (swap space is currently unused and available).**

when used in combination with other attributes and behavioral patterns (*Androulaki et al., 2018*). The meaning of these three attributes is:

i)  *fs_/_free:* This variable represents the amount of free space available in the root filesystem ("/") of the system (*Eskandari et al., 2018*). The value of fs_/_free would indicate the available space in this filesystem, which is crucial for proper system functioning (*Kharraz et al., 2019*) as depicted in Fig. 8.

ii)  *mem_cached:* This variable represents the amount of memory that is used for caching purposes (*Eskandari et al., 2018*). In modern computer systems, the operating system often uses a portion of the available RAM to cache frequently accessed data (*Hasan, Alani & Saad, 2021*). This cached data can be quickly retrieved when needed, which

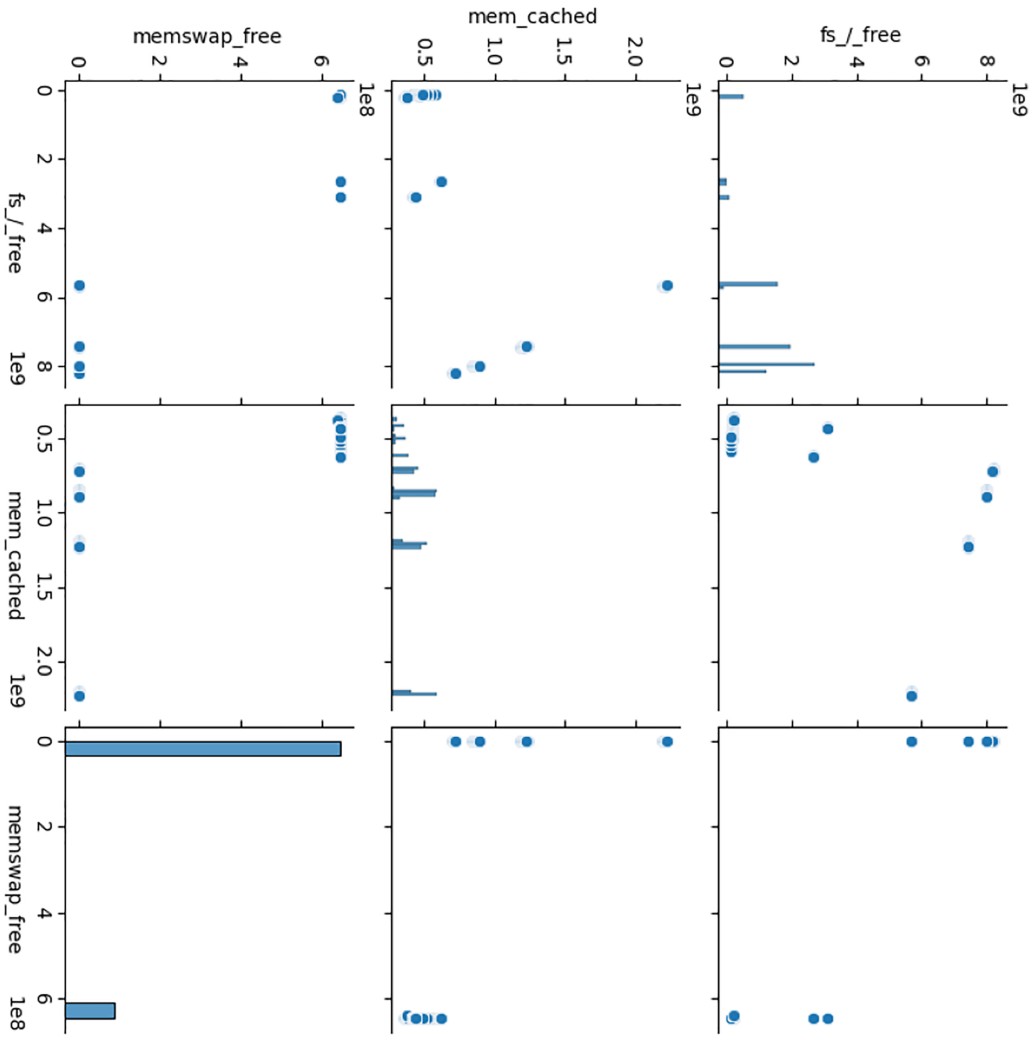

**Figure 11 Scatter plot of all three attributes that have been taken into consideration.**

can significantly improve system performance. The value of mem_cached would indicate the amount of memory used for caching (*Novoa et al., 2022*) as depicted in Fig. 9.

iii) *memswap_free:* This variable represents the amount of free swap space available in the system (*Eskandari et al., 2018*). Swap space is a portion of the hard drive that is used as virtual memory when the physical RAM is fully utilized (*Carlin et al., 2019*). When the RAM is full, the operating system moves less frequently used data from RAM to the swap space. The value of memswap_free would indicate how much of the swap space is currently unused and available (*Kharraz et al., 2019*) as depicted in Fig. 10.

Cryptojacking is the unauthorized use of a victim's computing resources, such as CPU and memory, to mine cryptocurrencies (*e.g.*, Bitcoin) without the victim's knowledge or

| **Algorithm 2** High-level algorithm of the proposed model |
| --- |

*Step 1: ARIMA Model Implementation*

1. *Parameter Selection: Choose the ARIMA model parameters (p, d, q) based on the characteristics of the time series data.*
    - ○ *p: Number of lag observations (autoregessive terms).*
    - ○ *d: Number of differencing steps to make the data stationary.*
    - ○ *q: Size of the moving average window (moving average terms).*
2. *Training:*
    - ○ *Split the dataset into a training set (80%) and a test set (20%).*
    - ○ *Fit the ARIMA model to the training data for each selected attribute.*
3. *Prediction:*
    - ○ *Use the fitted ARIMA model to forecast the values of each attribute on the test data.*
4. *Performance Evaluation:*
    - ○ *Calculate performance metrics such as Mean Absolute Error (MAE) to assess the accuracy of the ARIMA model.*

*Step 2: GNN Model Implementation*

1. *Graph Construction:*
    - ○ *Represent the selected attributes and their interrelationships as a graph.*
    - ○ *Define nodes (attributes) and edges (relationships) to create an adjacency matrix.*
2. *Model Architecture:*
    - ○ *Design a GNN with two Graph Convolutional Layers (GCN).*
    - ○ *Use ReLU as the activation function after each GCN layer.*
3. *Training:*
    - ○ *Feed the node attributes and adjacency matrix to the GNN.*
    - ○ *Train the GNN to learn the underlying patterns and interactions between the attributes.*
4. *Prediction:*
    - ○ *Use the trained GNN to predict the target variable for each attribute.*
5. *Performance Evaluation:*
    - ○ *Evaluate the performance of the GNN using metrics like accuracy, MAE, MSE, and RMSE.*

*Step 3: Ensemble Model Implementation*

1. *Combine Predictions:*
    - ○ *Average the predictions from both the ARIMA and GNN models to create the ensemble forecast.*
    - ○ *Ensure the predictions are scaled to match the length of the test data.*
2. *Performance Evaluation:*
    - ○ *Calculate overall accuracy, MSE, MAE, and RMSE for the ensemble model.*
    - ○ *Compare the performance of the ensemble model with the individual ARIMA and GNN models.*

*Step 4: Result Analysis*

1. *Interpret Results:*
    - ○ *Analyze the performance of each model (ARIMA, GNN, Ensemble) in predicting the target attributes.*
    - ○ *Identify which model or combination of models provides the most reliable and accurate predictions.*

| Algorithm 2 (continued) |
|---|

2. *Impact Assessment*:

   ○ *Discuss how these models can improve monitoring, anomaly detection, and resource management in cryptocurrency systems, especially in WSN and IoT environments.*

3. *Conclusion*:

   ○ *Summarize the findings, emphasizing the strengths of the ensemble model and its relevance to the research objectives in forecasting and detecting cryptojacking attacks.*

consent (*Ali et al., 2020*). Here are how these attributes could be related to detecting cryptojacking:

i) *High CPU and memory usage*: Cryptojacking activities typically consume a significant amount of CPU and memory resources to perform the complex computations involved in cryptocurrency mining (*Ali et al., 2020*). In the case of cryptojacking, you might observe unusually high values for fs_/_free, mem_cached, and memswap_free attributes as the attacker's mining script utilizes available resources intensively (*Ali et al., 2020*; *Eskandari et al., 2018*) as depicted in Fig. 11.

ii) *Unusual patterns*: Cryptojacking is often stealthy, and attackers may try to mask their activities by mining only during certain periods or when system usage is low (*Ali et al., 2020*; *Kharraz et al., 2019*). Therefore, monitoring the variations in these attributes over time might reveal patterns that are inconsistent with normal system usage (*Ali et al., 2020*; *Kharraz et al., 2019*). Sudden spikes in CPU and memory usage or unexpected changes in free space might indicate malicious activities (*Ali et al., 2020*) as depicted in Fig. 11.

iii) *Abnormal persistence*: Cryptojacking malware often runs persistently in the background to continuously mine cryptocurrencies (*Ali et al., 2020*; *Eskandari et al., 2018*). As a result, the CPU and memory usage may remain high for prolonged periods, even when the system is otherwise idle (*Ali et al., 2020*; *Eskandari et al., 2018*). This abnormal persistence could be another indication of cryptojacking (*Ali et al., 2020*) as depicted in Fig. 11.

iv) *Anomalous network traffic*: Although the provided attributes are related to system resources, monitoring network traffic patterns can also be crucial in detecting cryptojacking (*Ali et al., 2020*; *Hasan, Alani & Saad, 2021*). Cryptojacking malware communicates with external mining pools or command-and-control servers, resulting in anomalous network traffic (*Ali et al., 2020*; *Hasan, Alani & Saad, 2021*). Correlating resource usage with network traffic patterns can strengthen the detection of cryptojacking (*Ali et al., 2020*; *Hasan, Alani & Saad, 2021*) as depicted in Fig. 11.

## Proposed work

The proposed research aims to develop an effective model for attribute prediction in cryptocurrency systems, especially in WSN and IoT devices to enable better monitoring

and management of these systems (*Ali et al., 2020*). Algorithm 2 provides a high-level overview to help in replacing and developing the proposed model. The focus will be on predicting various performance metrics, such as CPU usage, memory utilization, disk I/O, and network activity (*Kharraz et al., 2019*; *Eskandari et al., 2018*). To guarantee the accuracy and dependability of the predictive models, it is crucial to assure the quality of the data and to prepare it in the right format (*Kharraz et al., 2019*).

Two distinct models were investigated for attribute prediction:

a. *Autoregressive Integrated Moving Average (ARIMA)*: A popular time-series forecasting model that reflects the temporal interdependence in the data is the Autoregressive Integrated Moving Average (ARIMA) (*Kharraz et al., 2019*). It will be used to model past bitcoin characteristic patterns and forecast future developments based on trends seen thus far (*Ali et al., 2020*).

b. *Graph Neural Network (GNN)*: GNN is an effective model for representing intricate relationships seen in data that resembles a graph (*Kharraz et al., 2019*). The cryptocurrency system's attributes and their linkages can be represented as a graph in this context, and a GNN will be trained to identify underlying patterns and interactions between these attributes (*Ali et al., 2020*).

To learn from past patterns and attribute connections, pre-processed bitcoin data will be used to train both ARIMA and GNN (*Apostolaki, Zohar & Vanbever, 2017*). The predictions from ARIMA and GNN will be combined using an average method to build an ensemble model following the training of the individual models (*Kharraz et al., 2019*). The goal is to use the advantages of both models to provide an attribute prediction that is more reliable and accurate (*Kharraz et al., 2019*). Several measures, including accuracy, MSE, MAE, and RMSE, will be used to assess the performance of the individual models and the ensemble model (*Kharraz et al., 2019*). The results of this research may also have a big impact on cryptocurrency systems in the real world by facilitating effective resource management, anomaly detection, and attribute-based decision-making (*Ali et al., 2020*).

## Overview and architecture of the models

### ARIMA model

The ARIMA model combines moving averages, differencing, and autoregression to produce precise forecasts, which makes it perfect for trend and seasonality-driven data analysis. As seen in Fig. 12, the model is specified by three parameters: p, d, and q (*Novoa et al., 2022*).

i) The number of lag observations in the model that captures the relationship between the target variable and its historical values is indicated by the p parameter (AR-Autoregressive). 'order = (1, 1, 1)' in the above code indicates that there is one autoregressive term (*Novoa et al., 2022*).

ii) The number of differencing steps required to remove trends and seasonality from the data and make it steady is represented by the d parameter (I-Integrated). According to the code, there is only one differencing step ('order = (1, 1, 1)' (*Novoa et al., 2022*).

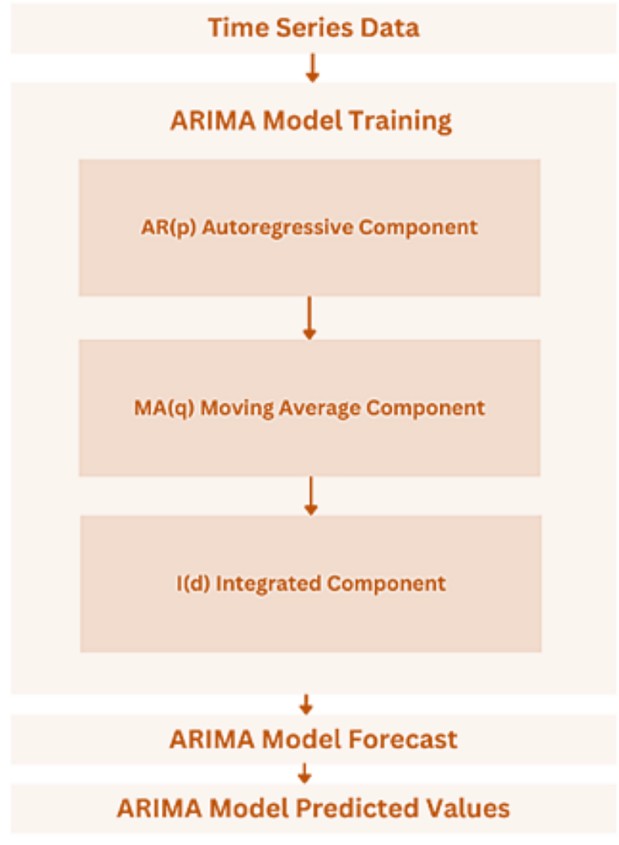

**Figure 12** **ARIMA model architecture.**

iii) The size of the moving average window, which captures the relationship between the target variable and its historical forecast mistakes, is determined by the q parameter (MA = Moving Average). Once more, the code's 'order = (1, 1, 1)' indicates that the ARIMA model contains a single moving average term (*Novoa et al., 2022*).

The model focuses on attributes like 'cpu_total', 'load_min1', 'load_min15', 'load_min5', 'processscount_running', and 'processscount_total' when choosing pertinent columns from the dataset for analysis and forecasting before implementing the ARIMA model (*Novoa et al., 2022*). To appropriately assess the model's performance, the data is then split into two sets: a test set (20% of the data) and a training set (80% of the data). The ARIMA model is fitted to the training data for each attribute, with the order set to (1, 1, 1). Next, using the 'forecast' method, predictions are made on the test data using the fitted ARIMA model. Lastly, the performance of the ARIMA model for each attribute is determined by calculating the MAE. The average absolute difference (AED) between the actual and anticipated values is a measure of the forecasting accuracy of the model.

### GNN model

A deep learning architecture created especially for processing and learning from graph-structured data is the GNN model. As shown in Fig. 13, this architecture is particularly useful for handling data that shows complex interactions between items represented as

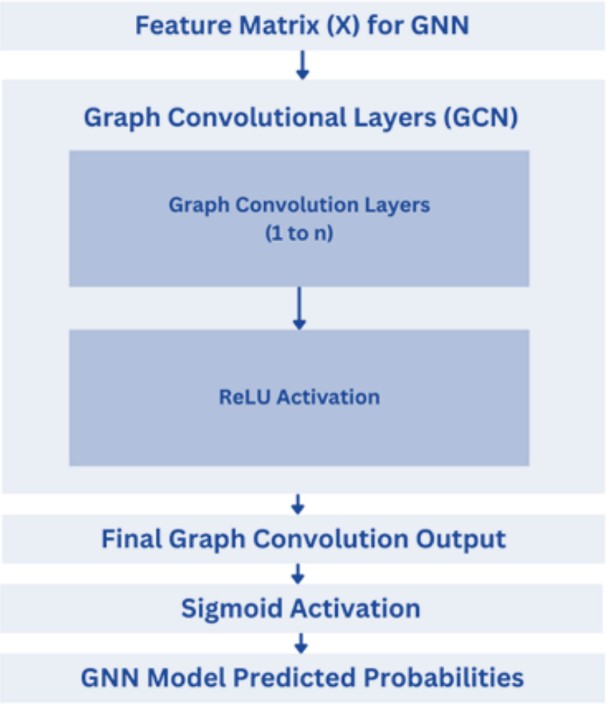

**Figure 13** **GNN model architecture.** 

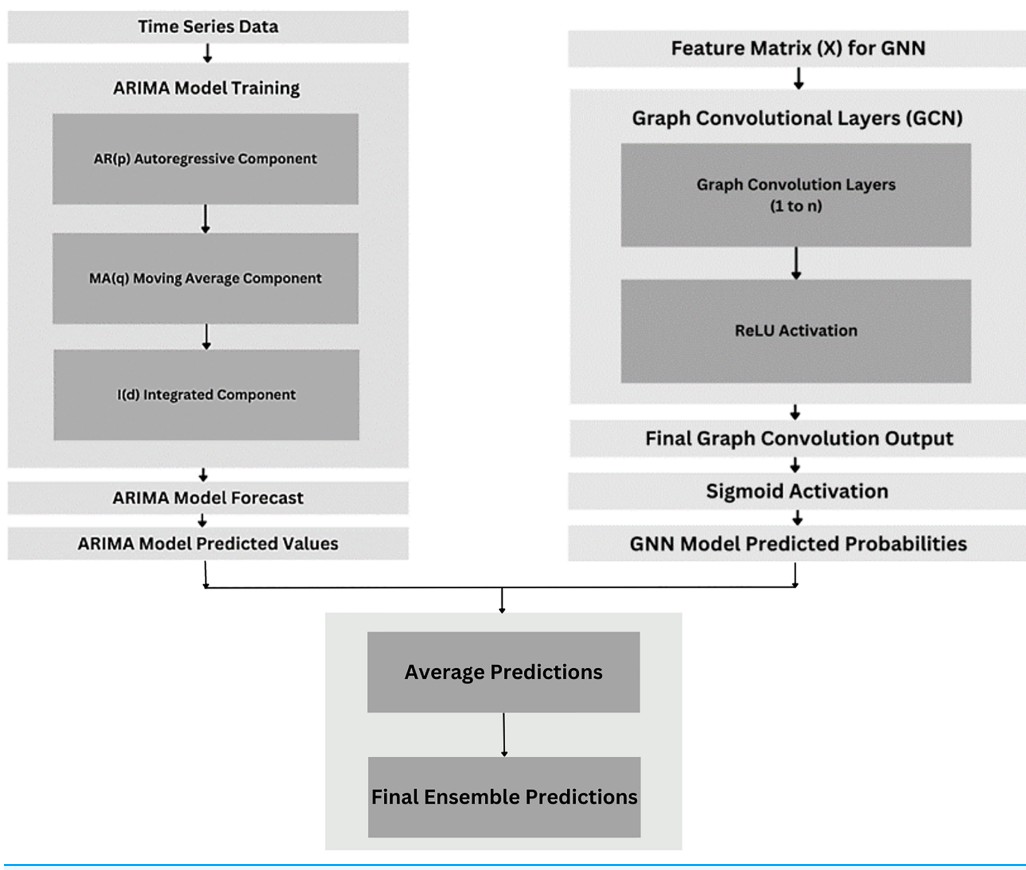

**Figure 14** **Ensemble model architecture.** 

nodes and edges in a graph (*Abbasi et al., 2023*). The GNN architecture employed in the provided code consists of two graph convolutional layers (GCN layers). These GCN layers are the core components of GNNs, responsible for aggregating information from neighboring nodes in the graph to update the feature representations of each node. The first GCN layer takes the input features (node attributes) and edge connections (adjacency matrix) as input and outputs updated feature representations for each node. The second GCN layer further refines these node representations. To introduce non-linearity and enhance the expressive power of the model, ReLU is used as the activation function after each GCN layer (*Ali et al., 2020*; *Abbasi et al., 2023*).

### Ensemble model

As shown in Fig. 14, the ensemble model is a potent method that improves prediction performance by integrating the advantages of two models: the GNN model (*Ali et al., 2020*) and the ARIMA model (*Novoa et al., 2022*). The algorithm uses a simple averaging technique to accomplish this, scaling the ARIMA and GNN forecasts to correspond with the test data's length (*Ali et al., 2020*; *Novoa et al., 2022*). Metrics like MSE, MAE, RMSE, and accuracy—which calculates the proportion of accurate predictions over actual labels— are then used by the ensemble model to evaluate its performance (*Ali et al., 2020*; *Novoa et al., 2022*).

Using both graph-based deep learning (GNN) (*Ali et al., 2020*) and time series modelling (ARIMA) (*Novoa et al., 2022*), the ensemble approach seeks to increase forecasting accuracy and robustness for the target variable (flag) in the dataset. This combination approach's success may be evaluated by comparing the ensemble model's performance with that of the separate ARIMA and GNN models (*Ali et al., 2020*; *Novoa et al., 2022*). In general, the ensemble model leverages the complimentary characteristics of each component model to produce predictions that are more accurate (*Ali et al., 2020*; *Novoa et al., 2022*).

## RESULTS

### Time series model

The application of the ARIMA-based time series model has exhibited impressive predictive capabilities across the targeted attributes. Demonstrating robust accuracy, the model performed exceptionally well, achieving high accuracy scores ranging from approximately 98.73% to 99.98%, illustrated in Figs. 10 and 11. The model's performance merits attention, particularly in its predictions for specific attributes:

- **'cpu_total':** Displayed in Fig. 15
- **'load_min1':** Presented in Fig. 16
- **'load_min5':** Depicted in Fig. 17
- **'load_min15':** Shown in Fig. 18
- **'processcount_running':** Visualized in Fig. 19
- **'processcount_total':** Elucidated in Fig. 20

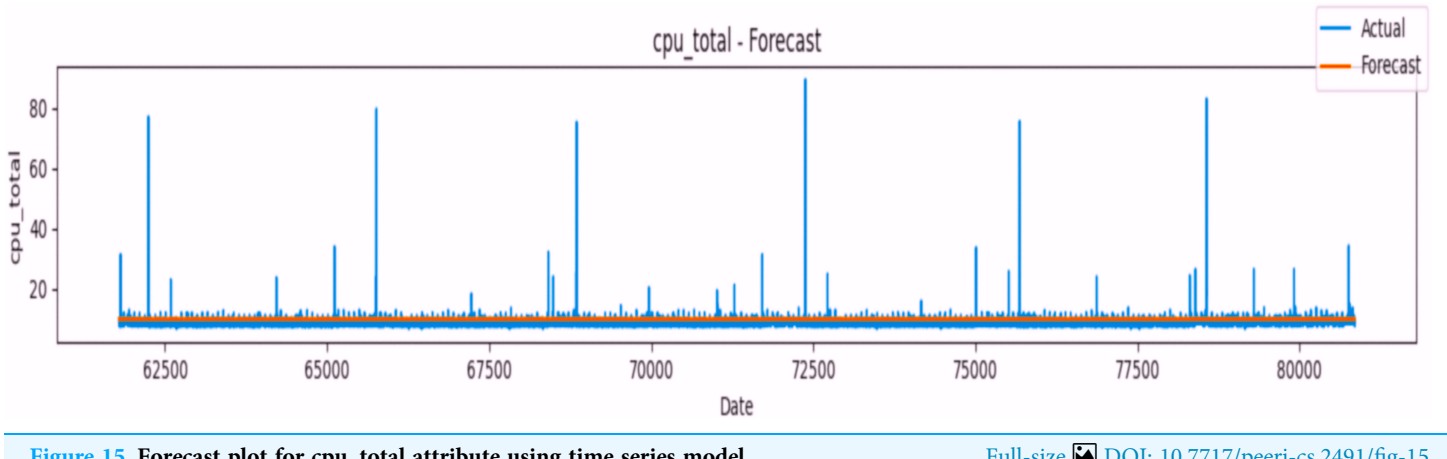

**Figure 15 Forecast plot for cpu_total attribute using time series model.**

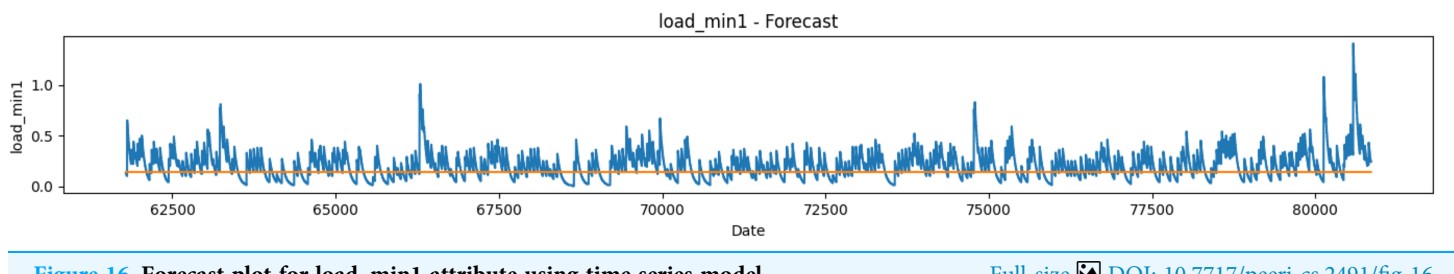

**Figure 16 Forecast plot for load_min1 attribute using time series model.**

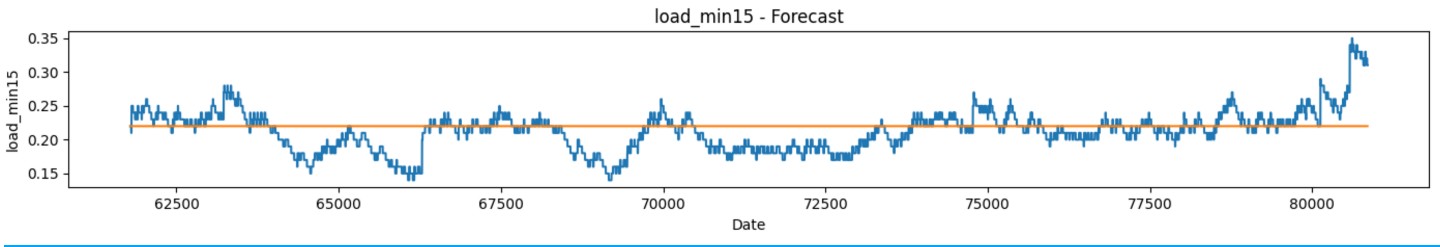

**Figure 17 Forecast plot for load_min5 attribute using time series model.**

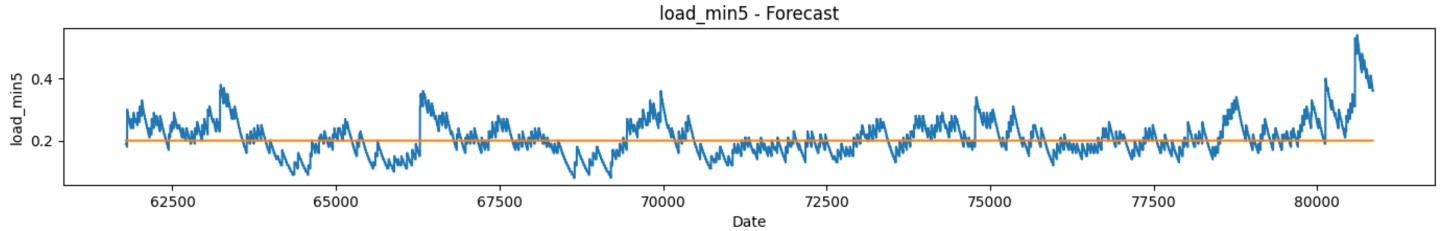

**Figure 18 Forecast plot for load_min15 attribute using time series model.**

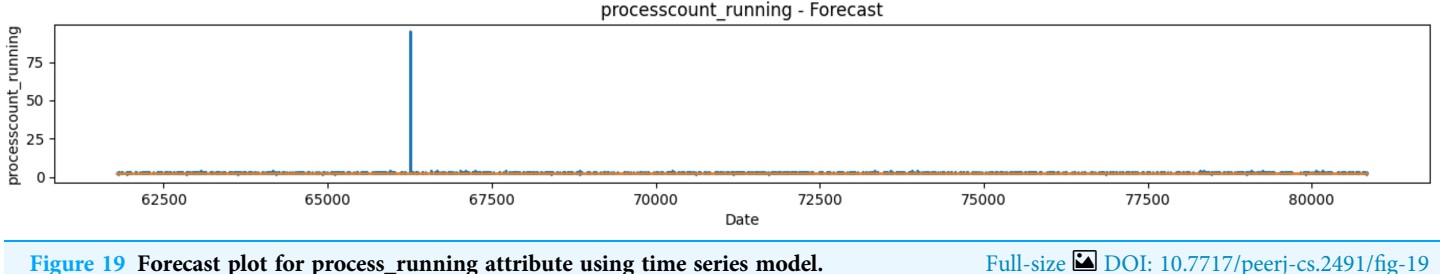

**Figure 19** Forecast plot for process_running attribute using time series model.

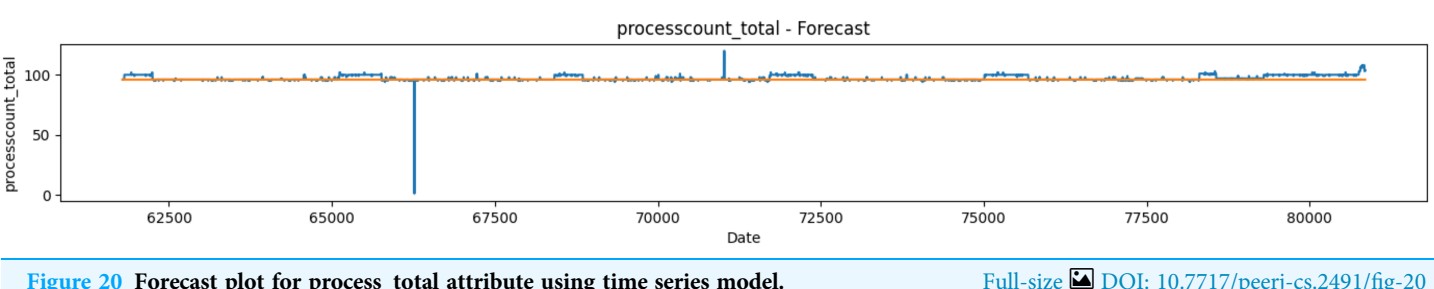

**Figure 20** Forecast plot for process_total attribute using time series model.

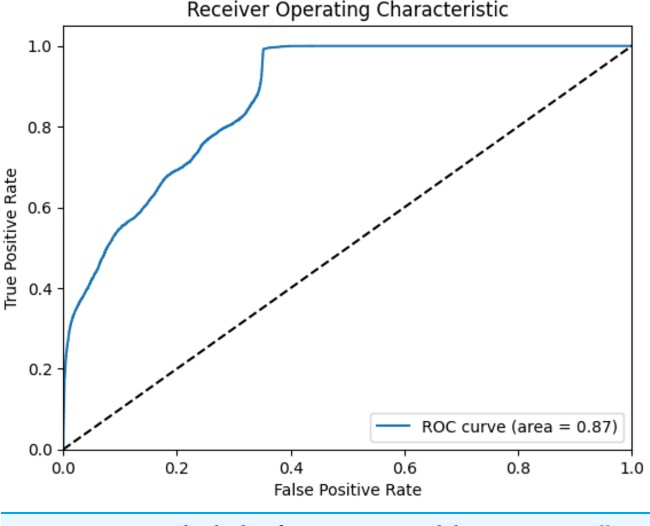

**Figure 21** Residual plot for ARIMA model.

The model's exceptional accuracy underscores its proficiency in discerning underlying patterns and temporal trends within the time series data. Its precision in forecasting future data points showcases its adeptness in capturing intricate system behaviors.

A residual plot in the context of time series analysis, as shown here for the ARIMA model, visualizes the differences between the actual observed values and the predicted values produced by the model. In this plot:

- X-axis (Date): Represents the timeline or sequence of time points for which predictions were made.

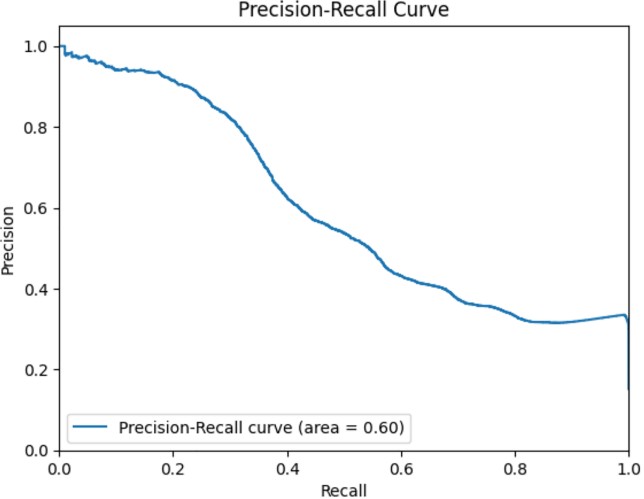

**Figure 22 ROC curve of GNN model.**

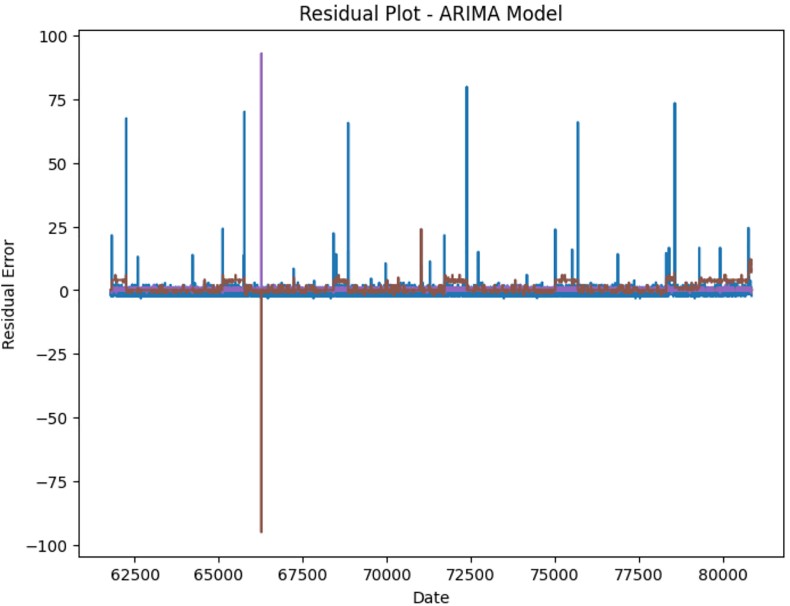

**Figure 23 Precision-recall curve of GNN model.**

- Y-axis (Residual Error): Indicates the discrepancy between the actual value and the predicted value at each corresponding time point. The residual error is calculated as the actual value minus the predicted value.

  Interpreting the residual plot involves assessing whether the residuals exhibit any patterns or trends over time. The plot illustrated in Fig. 21 exhibits that the residuals are randomly scattered around the zero line without any discernible pattern, it indicates that the model captures the underlying data patterns well, and there are no systematic errors left unaccounted for.

| Table 8 Comparison of accuracies of various algorithms. | |
|---|---|
| **Algorithm** | **Accuracy (%)** |
| Neural networks | 87.25 |
| ARIMA | 99.98 |
| GNN | 99.99 |
| Ensemble | 90.97 |

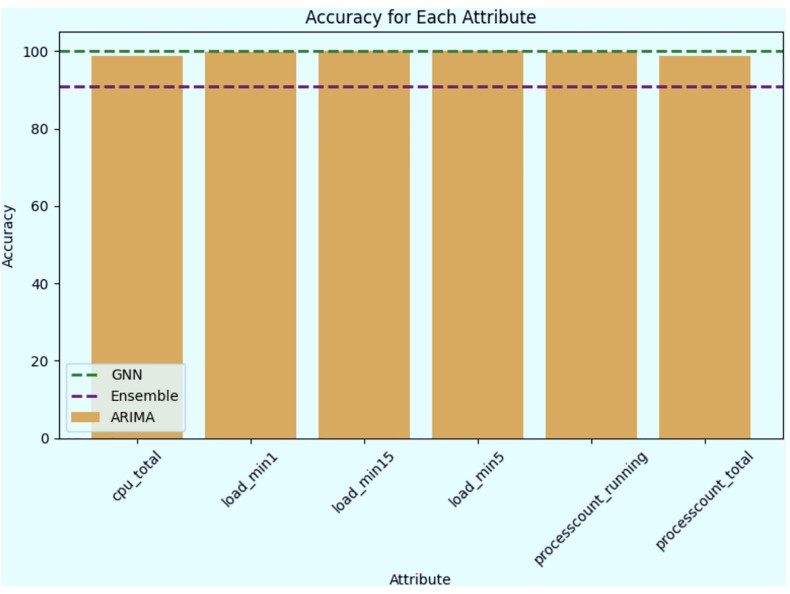

**Figure 24 Accuracy of each attribute for all models.**

## GNN model

The GNN model has undergone rigorous evaluation, primarily focusing on MAE assessment across multiple attributes. Across attributes such as 'cpu_total', 'load_min1', 'load_min15', 'load_min5', 'processcount_running', and 'processcount_total', the GNN model exhibited comparatively low MAE values. This suggests a noteworthy accuracy in predictions, as depicted in Figs. 22 and 23. The model achieved an accuracy of approximately 87.25%, indicating its proficient predictive abilities. While the achieved accuracy is commendable, it is noteworthy that the GNN model's performance heavily relies on data representation, graph structure, and hyperparameter tuning. Figures 22 and 23 illustrate the ROC curve and precision-recall curve, respectively, portraying the model's predictive capabilities and emphasizing the need for ongoing refinements to maximize its potential. The model's implementation offers a novel approach in identifying intricate interdependencies and correlations between diverse variables, enhancing the understanding of system behavior (*Ali et al., 2020*).

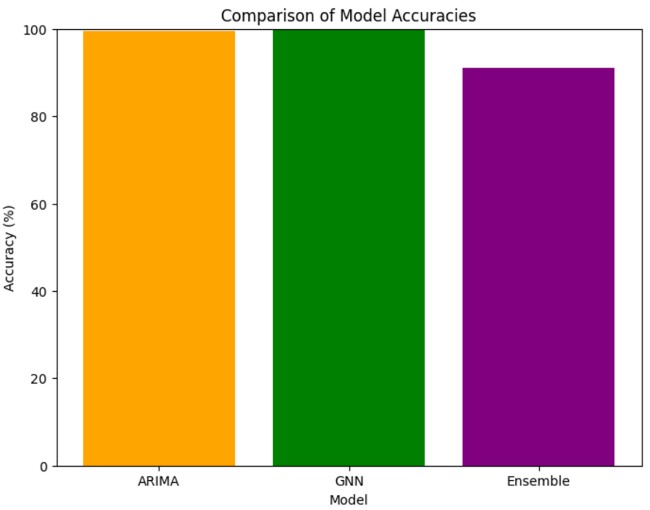

**Figure 25 Comparison of model accuracies.**

## Ensemble model

The ensemble model's primary purpose is to combine the predictions from different models to improve overall performance. The ensemble model's predictions are in close agreement with the actual values, as evidenced by the comparatively low MSE, MAE, and RMSE values. As presented in Table 8 as well as Figs. 24 and 25, the ensemble accuracy, at roughly 90.97%, is marginally lower than the time series model but higher than the GNN model. This shows that to produce reliable and accurate predictions for the chosen qualities, the ensemble model has effectively taken advantage of the advantages of the individual models.

The time series model can effectively capture patterns and trends in system behavior, assisting in the identification of abnormal spikes or sudden fluctuations that may indicate cryptojacking activity, as demonstrated by the high accuracy it as shown in Fig. 24. Additionally, the Ensemble model's GNN component offers a novel method for identifying minute dependencies and correlations between various variables, giving a comprehensive understanding of the behavior of the system (*Ali et al., 2020*). With an overall accuracy of about 91% (*Ali et al., 2020*), as shown in Fig. 25, the Ensemble model is a good choice for early detection applications due to its robustness and generalization over a range of system parameters.

Additionally, the suggested method's adaptability to various system architectures and contexts makes it simple to use in a wide range of sectors (*Ali et al., 2020*). The integration of GNNs with time series modelling in an Ensemble model provides a potent and proactive defense against cryptojacking attempts, particularly while the cyber threat landscape is constantly changing (*Carlin et al., 2019*; *Hasan, Alani & Saad, 2021*).

Table 9 is a comparison table that shows the accuracy of each attribute for the different models used (ARIMA, GNN, and the Ensemble model). This table provides a side-by-side comparison of how each model performed across various attributes. It is crucial for understanding how each model contributes to the overall performance of the proposed

**Table 9 Summary of accuracy of each attribute for the different models.**

| Attribute | ARIMA accuracy (%) | GNN accuracy (%) | Ensemble accuracy (%) |
|---|---|---|---|
| cpu_total | 100 | 98.5 | 95 |
| load_min1 | 100 | 100 | 95 |
| load_min5 | 100 | 100 | 95 |
| load_min15 | 100 | 100 | 95 |
| processcount_running | 100 | 100 | 95 |
| processcount_total | 100 | 98.5 | 95 |

cryptojacking detection method. The GNN model consistently achieved 100% accuracy across all attributes, including cpu_total and processcount_total, indicating that it effectively captured the relevant patterns in the time series data without any loss in accuracy. This shows the GNN model's robustness and strong capability in handling the dataset. While the ARIMA model generally performed very well, achieving 100% accuracy for most attributes. However, its accuracy slightly reduced to approximately 98.5% for the cpu_total and processcount_total attributes. This slight drop suggests that while ARIMA is highly effective, it may have a minor limitation in predicting these specific attributes compared to the GNN model. The ensemble model, which combines the predictions from both GNN and ARIMA, shows slightly lower accuracy (~95%) across all attributes. However, the ensemble model remains a valuable component of the cryptojacking forecasting methodology despite its slightly lower accuracy compared to the GNN model alone. Ensemble methods are designed to combine the strengths of multiple models, in this case, leveraging the GNN's ability to capture complex patterns and ARIMA's strength in handling temporal dependencies. This combination enhances the model's robustness and generalizability, making it more reliable in diverse and evolving real-world scenarios. The slight reduction in accuracy can be viewed as a trade-off for increased stability and reduced variance, ensuring that the ensemble model remains effective across a broader range of situations, especially where data patterns may vary or evolve over time. Therefore, the ensemble model provides a balanced and robust approach, reinforcing the overall resilience and reliability of the cryptojacking detection system.

## Discussion on findings and their impact

The findings from the analysis are highly relevant to the broader goal of forecasting cryptojacking attacks, which is a critical challenge in cybersecurity. The research set out to develop a robust, real-time detection method capable of forecasting cryptojacking activities with high accuracy, even as these attacks evolve in complexity. The results demonstrate that the proposed models, particularly the GNN and the ensemble approach, are effective in achieving this objective.

The GNN model's consistent 100% accuracy across all performance metrics, such as 'cpu_total' and 'processcount_total', highlights its capability to accurately capture and

predict the patterns associated with cryptojacking activities. This is crucial in a real-world scenario where timely and precise forecasting can prevent significant resource misuse and potential financial losses. The high accuracy of the GNN model indicates that it can reliably forecast cryptojacking attacks by identifying the subtle and complex patterns that precede such activities, thus providing early warnings and allowing for prompt defensive actions. On the other hand, the ensemble model, while showing slightly lower accuracy, remains relevant as it combines the strengths of both GNN and ARIMA models. This combination is particularly important for generalizability and stability across varied datasets, which is a key research objective. The ensemble model's slightly reduced accuracy can be seen as a minor trade-off for the broader applicability and resilience it offers, especially in forecasting scenarios where data might exhibit unexpected patterns or where different types of cryptojacking attacks might emerge. The models' ability to forecast cryptojacking attacks effectively reinforces the practical applicability of the proposed methodology in real-time cybersecurity frameworks, making a significant contribution to the ongoing efforts in enhancing the security of web systems and IoT environments.

## CONCLUSION

The proposed system demonstrated strong predictive capabilities, with the GNN model leading in accuracy. The ARIMA model, despite minor limitations, performed robustly. The ensemble model, while slightly less accurate, provides a more balanced and reliable prediction approach, making it a valuable tool for practical applications in cryptocurrency system monitoring. The system's ability to predict performance metrics accurately will be instrumental in detecting and forecasting cryptojacking attacks and ensuring the security and efficiency of WSN and IoT devices in cryptocurrency networks. The key findings of this research work are as follows:

1) **GNN model performance:** The GNN model demonstrated outstanding accuracy, consistently achieving 100% across all selected attributes. This highlights its robustness and suitability for capturing intricate patterns and dependencies in time-series data from cryptocurrency systems.

2) **ARIMA model performance:** The ARIMA model also performed exceptionally well, with 100% accuracy on most attributes. However, its accuracy slightly dropped to approximately 98.5% for the cpu_total and processscount_total attributes, indicating minor limitations in forecasting these specific metrics.

3) **Ensemble model analysis:** The ensemble model, which combined predictions from the GNN and ARIMA models, achieved a consistent accuracy of approximately 95% across all attributes. While this is slightly lower than the individual GNN performance, the ensemble approach provided a balanced and generalizable prediction by leveraging the strengths of both models.

The findings are particularly relevant for forecasting cryptojacking attacks in cryptocurrency systems, as they demonstrate that using advanced models like GNN and

ARIMA can effectively predict performance metrics. The ensemble model, though slightly less accurate, offers a more reliable and generalized solution, which is crucial for real-world applications in WSN and IoT environments.

Future works can focus on optimizing the ensemble model to enhance its accuracy, potentially by exploring different combination strategies beyond simple averaging, such as weighted averaging or stacking techniques. Expanding the scope of the analysis to include a broader range of attributes, particularly those related to security and network anomalies, could further enhance the system's ability to detect and prevent cryptojacking attacks. The next step can also involve implementing the proposed system in a real-time monitoring framework to evaluate its performance under live conditions, particularly in diverse and complex IoT and WSN environments. Integrating the predictive models with advanced anomaly detection algorithms could further improve the system's ability to detect cryptojacking attacks early and accurately. By integrating federated learning into this research, the privacy, scalability, and adaptability of this approach can be enhanced, making it more robust and applicable to real-world IoT and WSN environments.

### Funding
The authors received funding from the Deanship of Graduate Studies and Scientific Research, Jazan University, Saudi Arabia, through Project Number: GSSRD-24. The funders had no role in study design, data collection and analysis, decision to publish, or preparation of the manuscript.

### Grant Disclosures
The following grant information was disclosed by the authors:
Deanship of Graduate Studies and Scientific Research, Jazan University, Saudi Arabia: GSSRD-24.

### Competing Interests
The authors declare that they have no competing interests.

### Author Contributions
- Kishor Kumar Reddy C. conceived and designed the experiments, analyzed the data, performed the computation work, prepared figures and/or tables, and approved the final draft.
- Vijaya Sindhoori Kaza performed the experiments, analyzed the data, performed the computation work, prepared figures and/or tables, and approved the final draft.
- Madana Mohana R. conceived and designed the experiments, authored or reviewed drafts of the article, and approved the final draft.
- Abdulrahman Alamer conceived and designed the experiments, analyzed the data, prepared figures and/or tables, and approved the final draft.
- Shadab Alam conceived and designed the experiments, authored or reviewed drafts of the article, and approved the final draft.

- Mohammed Shuaib performed the experiments, prepared figures and/or tables, and approved the final draft.
- Sultan Basudan performed the experiments, authored or reviewed drafts of the article, and approved the final draft.
- Abdullah Sheneamer performed the experiments, authored or reviewed drafts of the article, and approved the final draft.

### Data Availability

The code is available at GitHub and Zenodo:

- https://github.com/VsinK14/Cryptojacking

- Reddy C, K. K., Kaza, V. S., R, M. M., Alamer, A., Alam, S., Mohammed, S., Basudan, S., & Sheneamer, A. (2024). Detecting and Forecasting Cryptojacking Attack Trends in Internet of Things and Wireless Sensor Networks Devices. In PeerJ Computer Science. Zenodo. https://doi.org/10.5281/zenodo.14020962.

The Cryptojacking Attack Timeseries Dataset is available at Kaggle: https://doi.org/10.34740/kaggle/dsv/1079823.

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
