# Peer review of "Detecting and forecasting cryptojacking attack trends in Internet of Things and wireless sensor networks devices"

_PeerJ Computer Science, doi:10.7717/peerj-cs.2491_

## Round 0.1 · original submission · Major Revisions

After the first round of review, based on the the comments from the reviewers, a “Major Revision” is recommended. Authors should consider the following comments.

1. Add more figures and diagrams to clarify the methodology and presented results.
2. Clearly state the contribution of the presented work or perhaps to compare the performance of this work with other published works.
3. Enhance the validation of the results.

Reviewer 1 ·

Basic reporting

The topic of the paper is a topic worth discussion.
However, if the references are related to forecasting of crypto jacking, it would be relevant to the proposed work.
The tables analyzed in the reference summary like Table1, 2 to 7 need to be explained more as to its relevance to the proposed work and its outcome.
The features selected with the need for such a selection is to be explained clearly.

Experimental design

The experimental results are relevant.
Each of the output figures have to be explained in terms o fits advantages to forecasting any future crypto jacking events.
The ensemble method proposed has been proved to have a reduced accuracy when compared to the ARIMA and GNN. How then do the authors justify that the proposed work is an improvement?

Validity of the findings

Various findings have been given.

However, the paper title suggests "forecasting" of a crypto jacking attack.
The authors have interchanged the words detection, prediction and forecasting. It would be suitable if same word is used.
Comparison with existing methods would enhance the value of the findings.

Each findings have to be explained on terms of its relevance to forecasting of the crypto jacking attack.

Additional comments

The research discussion is recent .
The overall grammar has to be checked and terms like "we" can be avoided in the paper.

Reviewer 2 ·

Basic reporting

The study of “Forecasting cryptojacking attack trends in Internet of Things using machine learning and deep learning techniques" has involved a lot of work from the authors.
The manuscript has been presented in an attractive manner.
There are recent references provided.
Need to separate Introduction and Literature Survey sections
Figures are relevant, high quality and labeled
However, there are a few areas where authors can make the manuscript better.

Experimental design

Research within the Scope of the journal.
Extensive research conducted in accordance with the technical and standards
A clearly stated, pertinent, and significant research question.
Techniques those are adequately detailed and informed

Validity of the findings

The underlying data are all available and are controlled, reliable, and statistically sound.
Well-stated conclusions that relate to the original research topic but need to add the performance of the proposed system and future work
Need to add Main findings under the results section

Additional comments

Below are some suggestions that will improve the manuscript and benefit a wide range of readers.

1. Authors should clearly specify the significance of Forecasting cryptojacking attack trends in Internet of Things in Introduction section
2. Correct the values in Table 5 and 6
3. If possible separate the introduction into introduction and Literature Survey sections.
4. Add Challenges and Objectives into the Literature Survey sections
5. Please discuss the Time Series Analysis after the line No. 307
6. Line No. 334, incorrect point number mentioned
7. Line No. 348, incorrect point number mentioned, Please check for whole article
8. For figure 23, if possible please add the table for Comparison of accuracy of Each Attribute for All Models
9. Under the results section, add the Main findings
10. In Conclusion section add the performance of the proposed system and future work

Reviewer 3 ·

Basic reporting

This research introduces a method that synergizes time series analysis techniques with graph neural networks to predict cryptojacking attack trends specifically in the context of wireless sensor networks and IoT. It was focused on early detection and anticipatory insights into emerging attack patterns within these interconnected ecosystems.

Here are a few comments and suggestions to improve the quality of the research:
1- Lack of novelty in the research.
2- The Abstract is generic about the developed model. The explanation does not reflect the work has been done for this research. It spent more sentences showing the results of the tested models.
3- The limitation and challenges presented in the introduction can be shown in a better way instead of just listing them and mentioned the corresponded research. It needs more explanation or presented in a proper way to connected it well with the context. paragraphs in introduction need to be well connected and have good smooth transitions.
4- The structure of the research. I’m not sure if this is the final structure of not. It needs to be organized in a better way. Listing the summarization of the tables without showing the tables until the end is very confusing.
5- Do we need all the presented tables? Also, do we need of the attributes and the entries shown in the tables? Where did we use all these information in the research paper?
6- The graphs must be in high-resolution
7- Some vocabularies need to be changed to the commonly used terms in research papers. Avoid uncommon vocabularies when possible.


Recommendation: Reject. It needs a novel solution. Also, it needs major improvements in the paper’s structure and the flow.

Experimental design

As above

Validity of the findings

As above

Additional comments

As above

Reviewer 4 ·

Basic reporting

No. Comment. OK.

Experimental design

OK for experimental design. The algorithm (Algorithm 1) can be presented in flowchart version also. Models (Time Series, ARIMA, GNN, Ensemble) can be compared with their features.

Validity of the findings

OK for results and findings. Conclusions can be stated in point-wise fashion to make it more clear along with future scope.

Reviewer 5 ·

Basic reporting

A thorough English check is required. Grammatical errors should be checked.
For comparative analysis elaborate how federated learning can be used in this context.

Experimental design

More figures should be added to give a description of the proposed technique.
Elaborate the proposed contribution in more detail.
More statistical analysis should be provided to support the results of proposed technique.

Validity of the findings

To validate the results, more discussion should be added.
Comparison with state of the art results should be added.
Figures quality should be improved. Please follow journal guidelines regarding the figures.

·

Basic reporting

Paper is written with clear English, proper thoughts, proper data etc.

Experimental design

Research questions, rigorous investigation, methods described - I found they are not up to the mark.

Validity of the findings

Validity of the findings - I am finding it difficult to comment on this.

Additional comments

Paper needs to be reframed again.

---

## Round 0.2 · accepted · Accept

Authors have addressed all the comments from the reviewers. Hence, this paper is recommended to accept in its current form.

Reviewer 1 ·

Basic reporting

The basic reports of literature survey and technical details are satisfied.

Experimental design

The experimental designs have been updated.

Validity of the findings

The GNN methodology and the accuracy improvement is well defined.

Additional comments

The paper has been updated for publication.

Reviewer 2 ·

Basic reporting

no comment

Experimental design

no comment

Validity of the findings

no comment

Additional comments

no comment

Reviewer 5 ·

Basic reporting

Authors have addressed all the comments.

Experimental design

Authors have addressed all the comments.

Validity of the findings

Authors have addressed all the comments.

Additional comments

Authors have addressed all the comments.